# Quantitative assessment of multiple pathogen exposure and immune dynamics at scale

Lusheng Song,[1] Femina Rauf,[1] Ching-Wen Hou,[1] Ji Qiu,[1] Vel Murugan,[1] Yunro Chung,[1,2] Huafang Lai,[1] Deborah Adam,[1] D. Mitchell Magee,[1] Guillermo Trivino Soto,[1] Milene Peterson,[1] Karen S. Anderson,[1,3] Stephen G. Rice,[1] Benjamin Readhead,[4] Jin G. Park,[1] Joshua LaBaer[1,5]

**ABSTRACT**   Serological responses reveal recent and historical exposure to pathogens, as well as the state of autoimmune and other chronic conditions, including cancer. Serological tests either assess one or a few antigens in many people, or multiple antigens for modest sample sizes. Here, we describe a multiplexed serology method that evaluates samples at the scale of thousands. This molecular epidemiology tool operates at a population scale sufficient to evaluate broadly how various infectious exposures or autoimmune responses affect health. The method employs full-length folded proteins, is quantitative over a wide dynamic range, and performs favorably compared with commercial clinical assays [severe acute respiratory syndrome coronavirus 2 chemiluminescent IgG II assay (Beckman) and Platelia SARS-CoV-2 total Ab enzyme-linked immunosorbent assay (Bio-Rad, California, USA)]. Responses to 39 bacteria species/strains and 99 viruses in 2,400 people were evaluated. Subjects with longitudinal data showed quantitative stability of response to all antigens over the 6-month time window, enabling the detection of intervening clinical events. We expect this highly adaptable method will find broad application in immune profile tracking.

**IMPORTANCE**   Serology reveals exposure to pathogens, as well as the state of autoimmune and other clinical conditions. It is used to evaluate individuals and their histories and as a public health tool to track epidemics. Employing a variety of formats, studies nearly always perform serology by testing response to only one or a few antigens. However, clinical outcomes of new infections also depend on which previous infections may have occurred. We developed a high-throughput serology method that evaluates responses to hundreds of antigens simultaneously. It can be used to evaluate thousands of samples at a time and provide a quantitative readout. This tool will enable doctors to monitor which pathogens an individual has been exposed to and how that changes in the future. Moreover, public health officials could track populations and look for infectious trends among large populations. Testing many potential antigens at a time may also aid in vaccine development.

**KEYWORDS**   multiplex assay, infectious disease surveillance, molecular diagnosis, human microbiome, virology, immunoassays, immunoserology, proteomics

The SARS-CoV-2 pandemic provides a potent reminder of the importance of tracking serological immune responses. Serology evaluates exposures and vaccinations in an individual as well as provides a public health tool to monitor disease transmission and immunity in communities. Employing a variety of formats, studies nearly always perform serology by testing response to only one or a few antigens. However, clinical outcomes of new infections also depend on exposures to prior pathogens. Immunity to some organisms can provide cross-species immunity to others; for example, meningitis

Address correspondence to Joshua LaBaer, jlabaer@asu.edu.

Lusheng Song and Femina Rauf contributed equally to this article. Author order was determined in order of experiment finalization and manuscript editing.

The authors declare no conflict of interest.

See the funding table on p. 21.

patients with immunity to meningococcal group B outer membrane vesicles also show a decrease in contracting gonorrhea (1). The converse also occurs when a prior infectious history may lead to worse outcomes with a different pathogen. Antibodies to one serotype of dengue virus can markedly increase the risk of severe dengue fever from another serotype (2). The range and magnitude of such relationships remain largely unknown. Although some anecdotal linkages have been noted, large-scale omics studies are not currently feasible to evaluate the effect that a prior immunological repertoire has on clinical outcomes with new infections and long-term health in general. Doing such studies requires scalable assays that evaluate immunity to many organisms in many people.

There is growing interest in developing multiplexed assays to assess humoral immunity, especially for using omics studies that exploit the power of systems analysis and augmented intelligence to understand broadly the effect of immune history on health. They will advance molecular epidemiology as well as serosurveillance. The ideal multiplexed assay would enable testing a broad range of organisms, such as all those relevant to particular area of study or concern, e.g., common respiratory pathogens, pathogens endemic to a particular geographical location, or common complicating infections in specific opportunistic scenarios. The assay should have a quantitative readout with a wide dynamic range and a sensitive limit of detection (LOD). Key among the features would be the ability to test many patients in order to improve discovery and statistical power.

A variety of multiplex serology assay formats for antigens of full-length proteins have been developed. These include methods that support moderate numbers of different antigens (10s), including color-marked bead-based fluorescence (3) and array-based chemiluminescence detection (4), and several methods that support thousands of antigens including protein arrays (5–7), which typically use fluorescent detection, and in solution assays using nucleic acid-tagged antigens, which use next generation sequencing as a readout (8–12). In several cases, the moderate antigen methods have been characterized in detail for their quantification, leading to their application in clinical studies; however, this has not happened yet for any of the assays that support high numbers of antigens (3, 4). Moreover, all of the above methods have thus far been limited to testing hundreds of samples or fewer (3–12).

We set out to develop a highly scalable clinical testing-compatible multiplexed serology method that could evaluate antibody immune responses to hundreds of antigens in thousands of patients simultaneously in 24 hours. This number would enable comprehensive testing in most clinical scenarios. We evaluated the method quantitatively and compared it to approved clinical assays. We conducted a serological survey at one of the largest public universities in the USA to measure the seroresponse to the immunodominant antigens of various common pathogens during the pandemic. This survey was carried out twice, with a 6-month interval between the two instances to assess the stability of our antibody profiles and their changes resulting from infections and vaccinations.

## RESULTS

### Multiplexed in-solution protein array

We sought to develop a quantitative multiplexed platform for assessing antibody responses from small volumes of serum that was compatible with clinical use. The multiplexed in-solution protein array (MISPA) platform employs a solution-based protein library, each protein antigen covalently linked to a unique DNA barcode (Fig. 1A). The use of barcodes, in lieu of attaching proteins to a surface, allowed antibody and target antigen to interact with solution phase kinetics. After mixing the barcoded protein library with serum or plasma (1 µL) and isolating antibody-bound antigens, barcodes from the bound antigens were amplified using PCR. During the amplification step, unique sample index codes were appended to the protein barcode, linking the sample

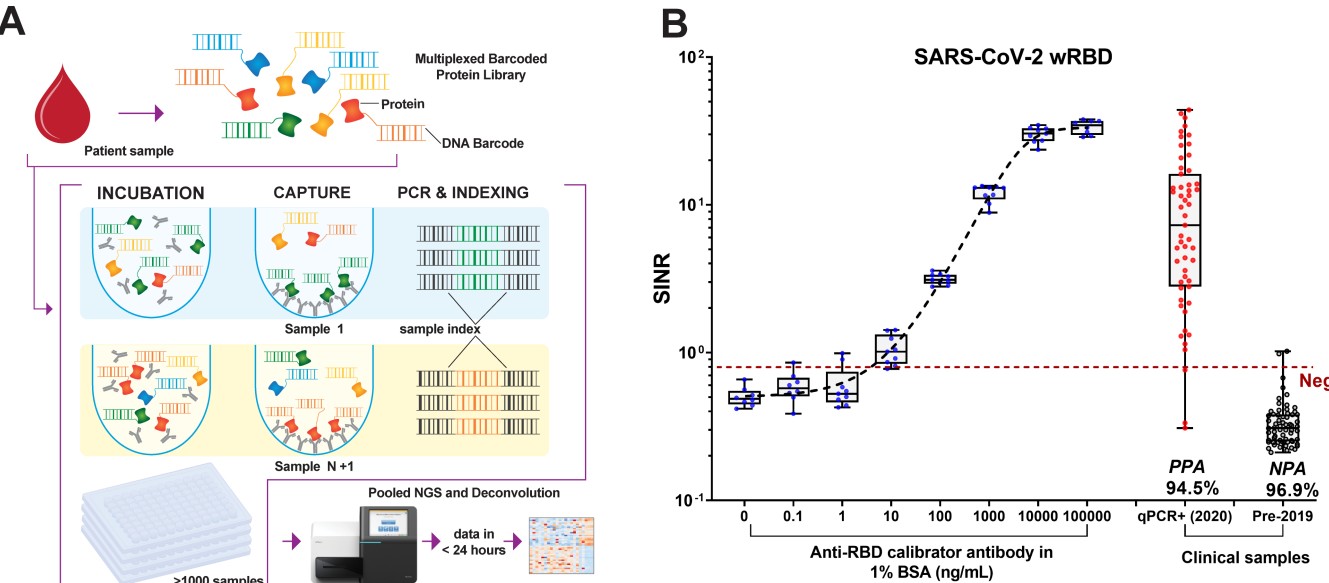

**FIG 1** Schematic workflow of multiplexed in-solution protein array (MISPA) and SARS-CoV-2 Wuhan strain, receptor-binding domain (wRBD) antibody calibration. (A) Schematic workflow for MISPA. (B) SARS-CoV-2 wRBD antibody calibration using MISPA. Serially diluted anti-receptor-binding domain antibodies were tested in nine technical replicates in MISPA, along with quantitative PCR + samples collected in 2020 and pre-2019 samples (box indicates median and 25th–75th percentiles, whiskers max and min samples). The mean value of pre-2019 sample plus 3× standard deviation was set as a seropositivity cutoff (red dashed line). Positive percent agreement (PPA) and negative percent agreement (NPA) with clinical diagnosis for serum samples were 94.5% and 96.9%, respectively. SINR, spike-in normalized ratio.

to its recognized antigens. The resulting PCR products were pooled and evaluated by next-generation sequencing (NGS).

DNA barcodes were covalently attached to HaloTagged target proteins through a chloroalkane ligand (Fig. 2: 1, click chemistry). The resulting ligand carried a universal sequence that was hybridized to an oligonucleotide with its complement and a 12-nt protein-specific barcode sequence that was subsequently filled in to produce a stable amplifiable double-stranded DNA barcode (Fig. 2: 2, barcode production). This flexible configuration accommodates thousands of unique protein-specific barcodes with sufficient dissimilarity between the sequences. Proteins were thus configured as chimeras with C-terminal HaloTag and 3xFLAG tags (Fig. 2: 3, protein expression).

The majority of proteins were produced using an *in vitro* transcription and translation system (IVTT). We also produced some protein targets using Expi293F cells when the IVTT-produced protein did not appear to be immunogenic. We used an in-gel fluorescence assay to assess the expression of individual proteins from both IVTT and Expi293F cells and a purified HaloTag protein as a positive control (+). The proteins were then barcoded with the barcode-linked haloalkane and isolated using anti-FLAG antibody-coated magnetic beads to remove unincorporated DNA barcodes (Fig. 2: 4, protein barcoding).

After incubation with serum and isolation on protein G beads, unique sample index codes were appended to the protein barcodes by PCR, and the final products were subjected to NGS (Fig. 2: 5, PCR and indexing). The amount of each antibody-bound barcoded protein in the sample could be assessed by counting the number of reads for PCR products with the linked barcodes and indexes.

We first tested a three-antigen MISPA assay that targeted two SARS-CoV-2 antigens of the original Wuhan strain, receptor-binding domain (wRBD) of the spike protein and nucleocapsid (NC), along with a negative control protein, the green fluorescent protein (GFP). The wRBD is a glycosylated protein with four disulfide bonds, and serum samples from COVID-19 patients showed only a weak response to IVTT-produced wRBD (13). Therefore, we expressed it in Expi293F cells. An in-gel fluorescence assay confirmed

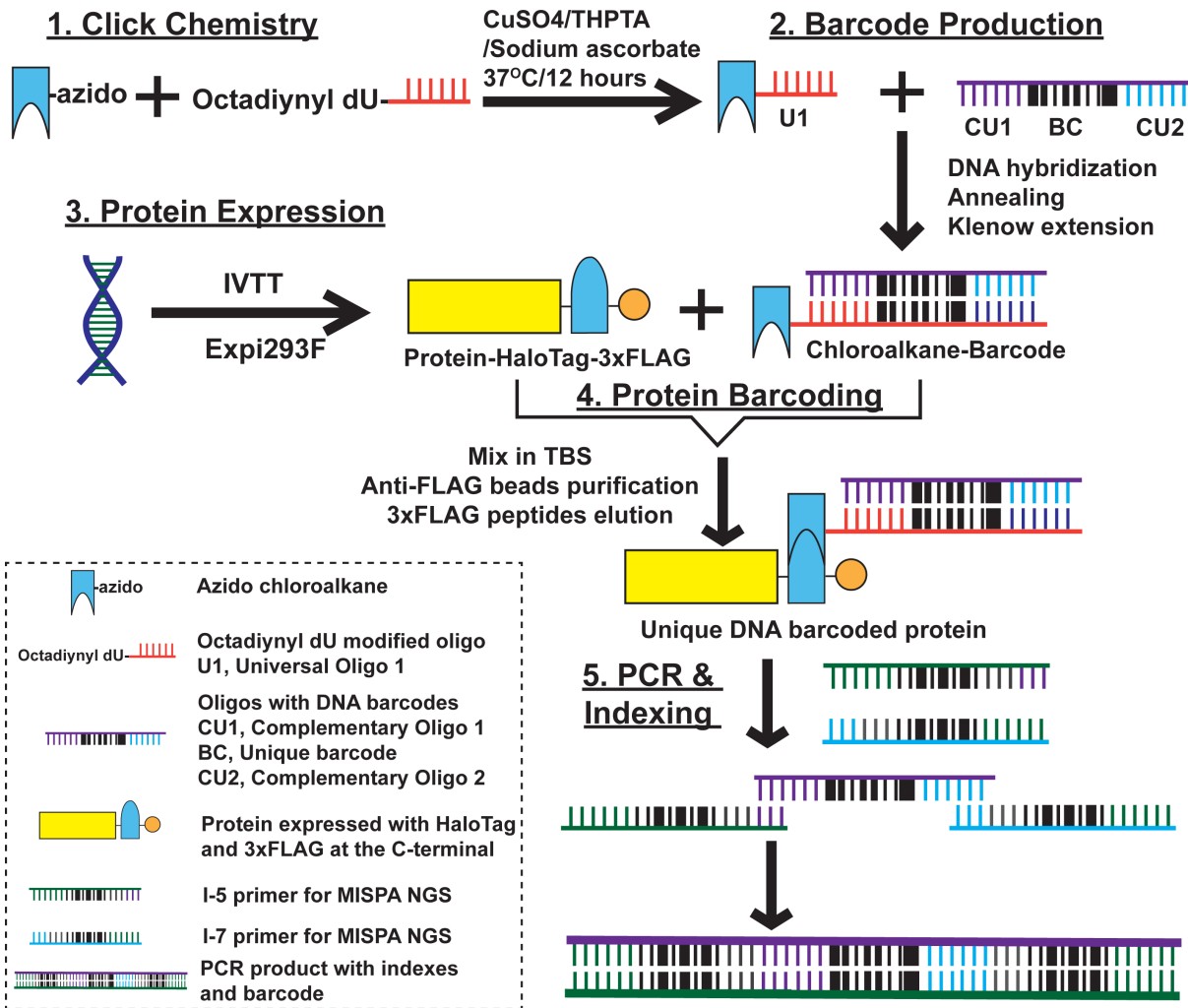

**FIG 2** Protein barcoding process. (1) Click chemistry: synthesized azido chloroalkane was incubated with octadiynyl dU modified oligo [universal oligo 1 (U1)], along with CuSO4, Tris(3-hydroxypropyltriazolylmethyl)amine, and sodium ascorbate; (2) barcode (BC) production: chloroalkane linked U1 was hybridized with oligo with DNA barcodes [complementary of U1 (CU1), unique barcode (BC), and complementary of universal oligo 2 (CU2)] and the annealed DNA was extended using Klenow enzyme; (3) protein expression: proteins were expressed using cell-free *in vitro* transcription and translation (IVTT) coupled solution or Expi293F human cell expression system; (4) protein barcoding: the crude protein expression solution was mixed with equal volume of chloroalkane ligand barcode to allow the covalent bonding between HaloTag and chloroalkane ligand; the barcoded protein was purified using anti-FLAG magnetic beads pull-down and eluted with 3xFLAG peptide; (5) PCR and indexing: sample-specific indexes were appended at both ends of the protein-specific barcode through PCR amplification, enabling up to 2,000 samples to be analyzed in the same experiment. The resulting barcodes for all samples were combined and assessed by NGS.

HaloTag functionality and polypeptide sizes (14) (Fig. S1A). The barcoded wRBD, NC, and GFP proteins were mixed to form an equimolar three-antigen MISPA library by assessing their product yield with PCR (Fig. S1B).

Three-antigen MISPA performance was assessed by analyzing 64 pre-2019 samples and 55 early 2020 samples collected from quantitative PCR (qPCR) test-confirmed COVID-19 patients. After mixing the MISPA library with patient samples, IgG antibody-bound antigens were captured by protein G beads, and the target antigen barcodes were amplified using PCR in the presence of spike-in control oligos (1 pM). After adding the sample index barcodes, the resulting PCR products were pooled and evaluated by NGS.

The number of sequence reads attributed to each sample index and protein barcode provided a quantifiable assessment of antibodies with a large dynamic range and little background. To normalize for the well-to-well variability in PCR efficiency, the raw antigen read counts for each sample were normalized by the corresponding spike-in

oligo read counts, i.e., a spike-in normalized ratio (SINR), before statistical analysis. A contrived pre-2019 sample was used as a negative control to determine seropositivity cutoffs [mean + 3× standard deviation (SD)] for wRBD and NC. The positive percent agreement (PPA) and negative percent agreement (NPA) with qPCR-verified COVID-19-positive and pre-2019 samples, respectively, for MISPA wRBD were 94.5% and 96.9%, respectively (Fig. 1B). The PPA for NC using MISPA was 90.9% and its NPA was 98.4% (Fig. S1C). Samples from some COVID-19 patients with negative MISPA test results were collected within a week of the symptom onset, likely before the development of an IgG antibody response. The LOD of wRBD on MISPA was assessed by analyzing a serially diluted monoclonal mouse anti-receptor-binding domain (RBD) antibody (0.1–100,000 ng/mL, 10×) to be 38.9 ng/mL [95% confidence interval (CI) 8.8–68.8 ng/mL] (Fig. S1D). The dynamic range for wRBD was from 38.9 to 100,000 ng/mL of anti-RBD (Fig. 1B).

## Microbial antigen MISPA

A key feature of the MISPA platform is the capacity to evaluate many antigens simultaneously. In addition to the two SARS-CoV-2 antigens, we added 141 microbial antigens that span 39 bacteria species/strains and 99 viruses (Table S1). Three autoantigens, p53, IFN-α2, and IFN-α4, were also included, as well as GFP, as a non-reactive protein control. The coronavirus antigens (SARS-CoV-2 and seasonal) were selected based on whole proteome studies on our protein microarrays, as well as the literature, that showed RBD and NC were the most seroreactive. We also included many respiratory pathogens, as well as other common pathogens of interest. For these other pathogens, where possible, we included the antigens from each that showed the most prevalent responses in our protein microarray studies. The RBD proteins from SARS-CoV-2 and three seasonal coronaviruses (229E, OC43, and HKU1) were expressed in the Expi293F cells. The remaining proteins were produced by IVTT (Table S1). All proteins expressed well and showed a dominant band representing the combined molecular weight of the antigen plus HaloTag (Fig. S2). Each protein was uniquely barcoded, and pre-test NGS evaluation of protein samples confirmed that each protein barcode was unique and there was no cross-contamination (Fig. 3A).

The microbial antigen MISPA library was prepared by mixing barcoded proteins using equal volumes. To demonstrate the appropriate stoichiometry, NGS of the final library revealed that the SINR for all antigens was higher than the negative control (no protein, mean + 3× SD). Signals were comparable among the proteins with the majority falling within two fold of the mean, 17.8 [8.3–29.0 (10%–90%)] (Fig. 3B). The LOD for wRBD when combined with 146 other antigens in the microbial MISPA library was 49.8 ng/mL (95% CI 9.1–90.5 ng/mL), which showed little change from the three-antigen assay, indicating that the presence of other antigens does not interfere with detection.

Reproducibility was assessed using 185 samples randomly selected from a serosurvey in Arizona State University (ASU) community on 1–3 March 2022 (SurveySpring22). The average coefficient of variation (CV) among all antigens was 23.7% ± 6.7% (Fig. 3C), with a trend toward higher CVs when signals were weak (i.e., near zero). Scatter plots of SINR values between independent MISPA runs for selected antigens [SARS-CoV-2 wRBD, NC, omicron receptor-binding domain (oRBD), EBNA-1, and NC proteins from seasonal coronaviruses] showed consistent results (Fig. 3D). A commercial pre-2019 mixed serum sample revealed a correlation of $R^2$ >0.99 between two independent runs (Fig. 3E). The use of different batches of protein library did not affect the quantitative outcome for either IVTT or Expi293F produced proteins. We prepared protein libraries, PL_147 in January 2023 and PL_184 in February 2023, both including SARS-CoV-2 wRBD. Eighty-one samples randomly selected from a serosurvey in the ASU community on 13–17 September 2021 (SurveyFall2021) were probed with both of the protein libraries, and the SARS-CoV-2 wRBD (Expi293F), SARS-CoV-2 NC, and EBNA-1 (IVTT) response revealed highly consistent responses in both protein libraries (Fig. 3F).

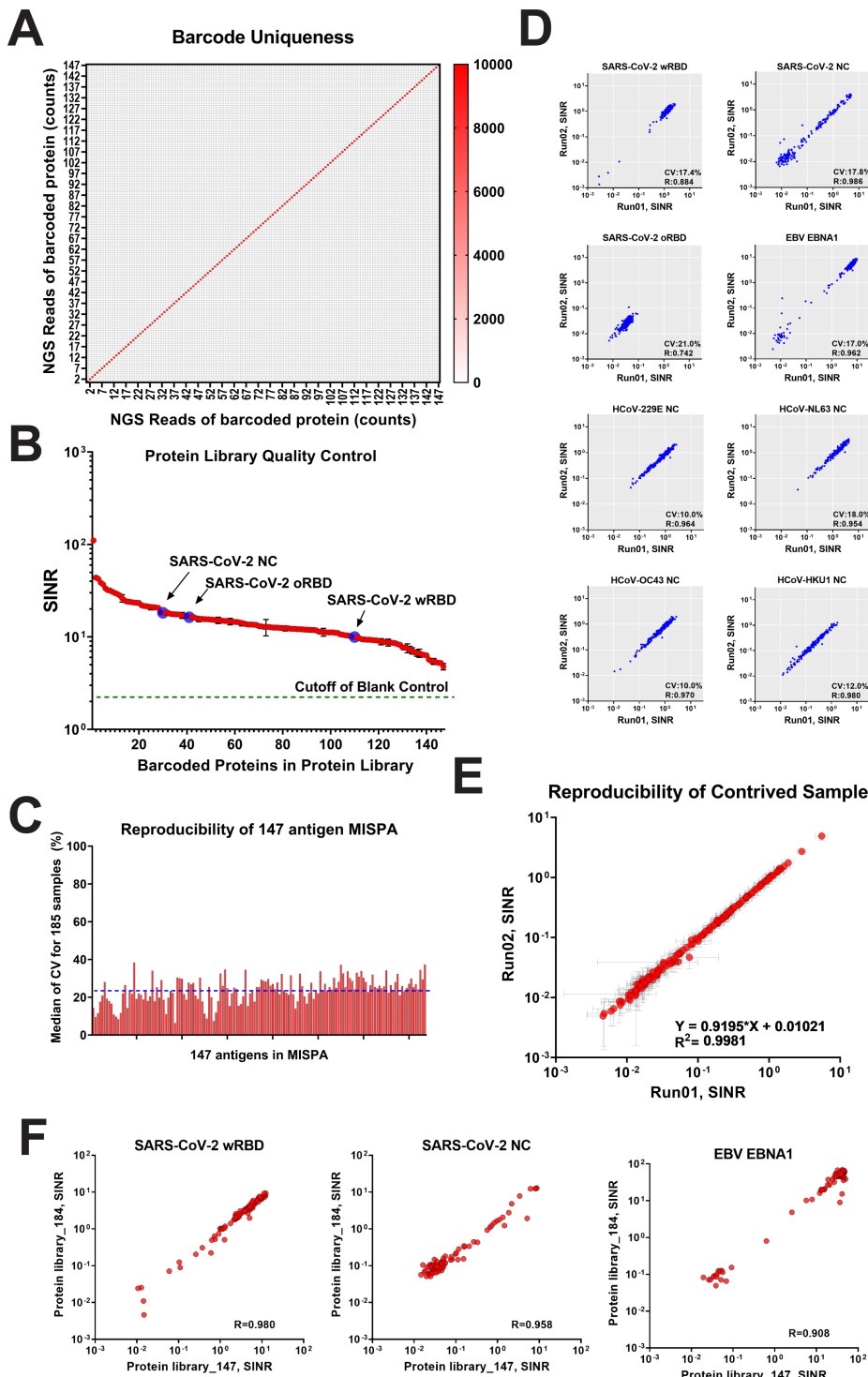

**FIG 3** Microbial MISPA assay quantification. (A) NGS analysis of the 147 individual barcoded proteins. The color scale indicates the raw count reads of NGS. (B) NGS analysis of multiplexed MISPA library with 147 barcoded proteins. Severe acute respiratory syndrome coronavirus 2 (SARS-CoV-2) wRBD, omicron receptor-binding domain (oRBD), and nucleocapsid (NC) are indicated by arrows. The cutoff value set by blank control [barcoding without protein, mean + 3× standard deviation (SD)] was used to determine the presence of individual barcoded proteins (dotted line). The mean spike-in normalized ratio (SINR) value for all proteins was 17.8 ± 13.0. (C) Bar plot of the reproducibility of 147-antigen MISPA. The coefficients of variance (CVs) for all 147 antigens had a median of 23.7% ± 6.7%. Sample order is from highest to lowest SINR value (left to right). (D) The reproducibility of selected antigens [SARS-CoV-2 wRBD, NC, oRBD, Epstein-Barr virus (EBV) EBNA1, and NC proteins

**FIG 3** (Continued)

of human coronavirus (HCoVs)] were plotted between two independently run MISPA assays on 185 samples. The CVs and linear regression $R^2$ values are listed in the lower right corner for each plot. (E) Reproducibility of a contrived sample. A commercial contrived pre-2019 pooled serum was analyzed for 147 antigens in MISPA. The responses of all 147 antigens were compared between two independent assays. The mean value was plotted as a red dot with whiskers indicating 1× SD. The linear regression equation and $R^2$ values are shown in the lower right corner. (F) Reproducibility of SARS-CoV-2 wRBD, NC, and EBNA-1 in two independent protein libraries (_147 and _184). The responses were compared between the same 81 samples. The linear regression $R^2$ values are shown in the lower right corner.

## Serosurvey using microbial antigen MISPA

Serosurveys were performed on 13–17 September 2021 (SurveyFall21, 1,064 participants) and on 1–3 March 2022 (SurveySpring22, 1,379 participants) to determine the rates of SARS-CoV-2 vaccination and natural infection within our generally healthy ASU community (Table 1). Students and employees at all ASU campuses were invited to participate through widely broadcast announcements. Of the 2,443 participants, 2,306 (94.4%) were unique to one of the two surveys. SARS-CoV-2 vaccination/infection status and other demographic information were self-reported (Table 2; Table S2).

We evaluated responses to SARS-CoV-2 antigens using MISPA and compared them to commercially available Food and Drug Administration (FDA) Emergency Use Authorization (EUA) assays (Table 2), including RBD (Beckman Access SARS-CoV-2 chemiluminescent IgG II assay) and NC [Bio-Rad Platelia SARS-CoV-2 total Ab enzyme-linked immunosorbent assay (ELISA) assay]. As expected from the highly reported frequency of vaccination, the self-reported vaccinated (Vax+) population for the fall and spring surveys approached 100% prevalence for anti-RBD antibodies [98.9% (MISPA)/92.4% (Beckman) and 98.9% (MISPA)/97.9% (Beckman), respectively]. Among those who reported a history of COVID-19 infection, we observed a prevalence of anti-NC antibody responses of 84.3% (MISPA)/56.9% (Bio-Rad, California, USA) for fall and 87.5% (MISPA)/72.0% (Bio-Rad) for spring. Overall, MISPA was comparable to or more sensitive than the corresponding commercial assays.

The responses for all antigens are shown as heatmaps (Fig. 4). Many antigens had low or undetectable signals in this population (e.g., human immunodeficiency viruses 1 and 2, adeno-associated virus, and yellow fever virus), whereas more than 96.3% (SurveyFall21) and 99.1% (SurveySpring22) of subjects had strong responses to wRBD, one of the most prevalent responses. There were 37 other microbial antigens (27.0%) that had strong responses in more than 90% of subjects, including SARS-CoV-2 and those from other respiratory viruses [e.g., human parainfluenza virus 3 (15), human respiratory syncytial virus (16, 17), seasonal coronavirus (18), influenza A virus (19) and influenza B virus, rhinovirus A (20), and human mastadenoviruses B–D (16, 17)], gastrointestinal viruses [e.g., enteroviruses A and B (21)], and pathogenic bacteria [e.g., *Staphylococcus aureus* (22), *Haemophilus influenzae* (23, 24), *Pseudomonas aeruginosa*, and *Klebsiella oxytoca*], all of which have been reported to have close to 100% seroprevalence in individuals more than 2 years old.

Except for antigens from the same genus [such as human papilloma viruses (HPVs) and enteroviruses], which share high sequence homology, antibody response to most antigens showed unique patterns in our population, as shown by a large least squares distance in their response correlation analysis (Fig. 4). The main driver for wRBD response was vaccination; it did not correlate with any other antigen except for oRBD, where the Pearson correlation coefficients were 0.80 and 0.81 for SurveyFall21 and SurveySpring22, respectively (Fig. 4). The correlation clustering patterns across all antigens were very similar between two surveys with a sum squared value of 0.859 ($P < 0.001$) (Fig. S3).

There was no apparent correlation among responses to wRBD or NC from SARS-CoV-2 and their counterparts from the seasonal coronaviruses (sCoV) (Pearson correlation coefficient <0.5) (Fig. 4). However, we noted that subjects within the fifth percentile of abundance for each of the sCoV antigens, as well as SARS-CoV-2 NC, had an

**TABLE 1** Demographic parameters for serosurveys in September 2021 and March 2022

| | Serosurvey in September 2021 (n = 1,064) | Serosurvey in March 2022 (n = 1,397) |
|---|---|---|
| Gender, n (%) | | |
| Female | 558 (52.44) | 794 (56.84) |
| Male | 467 (43.89) | 570 (40.8) |
| Others | 13 (1.22) | 22 (1.57) |
| Missing | 26 (2.44) | 11 (0.79) |
| Age, n (%) | | |
| 18–49 | 988 (92.86) | 1,186 (84.9) |
| ≥50 | 42 (3.95) | 184 (13.17) |
| Missing | 34 (3.2) | 27 (1.93) |
| Race, n (%) | | |
| White | 521 (48.97) | 656 (46.96) |
| Asian | 295 (27.73) | 393 (28.13) |
| Mixed | 51 (4.79) | 100 (7.16) |
| Black | 26 (2.44) | 37 (2.65) |
| Native | 14 (1.32) | 9 (0.64%) |
| Other | 105 (9.87) | 37 (2.65) |
| Prefer not to say | 9 (0.85) | 0 (0) |
| Missing | 27 (2.54) | 11 (0.79) |
| Vaccination status | | |
| Yes | 978 (91.92) | 1,323 (94.7) |
| No | 82 (7.71) | 61 (4.37) |
| Missing | 4 (0.38) | 13 (0.93) |
| Vaccine source | | |
| Pfizer | 510 (47.93) | 668 (47.82) |
| Moderna | 309 (29.04) | 443 (31.71) |
| Janssen | 94 (8.83) | 93 (6.66) |
| AstraZeneca | 46 (4.32) | 97 (6.94) |
| Attenuated virus | 12 (1.13) | 16 (1.14) |
| Missing | 93 (8.74) | 80 (5.73) |
| Previous COVID-19 infection | | |
| Yes | 205 (19.27) | 529 (37.87) |
| No | 857 (80.55) | 853 (61.06) |
| Missing | 2 (0.19) | 15 (1.07) |

unexpectedly high overlap between the subjects who demonstrated low sCoV and low SARS-CoV-2 NC antibody abundance (Fig. 5A and B). Further examination of the 17 subjects who were driving this enrichment revealed that they were significantly more likely to have self-reported a history of never having COVID-19 (false discovery rate [FDR] 1.20e-2, odds ratio 1.53, one-sided Fisher's exact test; Fig. 5C). The same findings were observed in a combined version of the data that included the earlier SurveyFall21 data (Table S3). We further compared the different response levels for the four sCoV antigens between those who reported having had COVID-19 and those who did not. The self-reported COVID-19 positives who were confirmed (SR+, confirmed with Bio-Rad Platelia SARS-CoV-2 total Ab ELISA assay anti-NC positive) were compared to the negatives (SR−, confirmed with Bio-Rad Platelia SARS-CoV-2 total Ab ELISA assay anti-NC negative) using the rank-sum test. Interestingly, the anti-NC responses for human coronavirus (HCoV)-229E and HCoV-NL63 (alpha coronavirus) were both significantly higher in SR+ (chi-square test $P$ value < 0.05), in both serosurveys, while there was no difference for anti-NC against the beta coronaviruses (HCoV-HKU1 and HCoV-OC43). One explanation could be antibody cross-reactivity; however, the NC protein from SARS-CoV-2 (a beta coronavirus) has higher sequence similarity to the NC proteins from the two seasonal beta coronaviruses (25) (Fig. S4). Nonetheless a similar observation

**TABLE 2** Prevalence of RBD and NC antibodies for SurveyFall21 and SurveySpring22[a]

| | Vaccination | | COVID-19 | |
|---|---|---|---|---|
| | Vax+ | Vax− | SR+ | SR− |
| Serosurvey September 2021 (*N* = 1,060) | | | | |
| | *n* = 978 (92.3%) | *n* = 82 (7.7%) | *n* = 204 (19.2%) | *n* = 856 (80.8%) |
| RBD | | | | |
| Beckman | 904 (92.4%) | 31 (37.8%) | 189 (92.6%) | 746 (87.1%) |
| MISPA | 967 (98.9%) | 54 (65.9%) | 204 (100.0%) | 817 (95.4%) |
| NC | | | | |
| Bio-Rad | 176 (18.0%) | 33 (40.2%) | 116 (56.9%) | 93 (10.9%) |
| MISPA | 307 (31.4%) | 49 (59.8%) | 172 (84.3%) | 184 (21.5%) |
| Serosurvey March 2022 (*N* = 1,379) | | | | |
| | *n* = 1,318 (95.6%) | *n* = 61 (4.4%) | *n* = 528 (38.3%) | *n* = 851 (61.7%) |
| RBD | | | | |
| Beckman | 1,290 (97.9%) | 39 (63.9%) | 516 (97.7%) | 813 (95.5%) |
| MISPA | 1,302 (98.8%) | 49 (80.3%) | 517 (97.9%) | 834 (98.0%) |
| NC | | | | |
| Bio-Rad | 500 (37.9%) | 37 (60.7%) | 380 (72.0%) | 157 (18.4%) |
| MISPA | 662 (50.2%) | 45 (73.8%) | 462 (87.5%) | 245 (28.8%) |

[a]Not included in this analysis were four samples that did not have vaccination or COVID-19 information. Vax+ and Vax− indicates vaccinated and not-vaccinated; SR+ and SR− indicates self-reported COVID-19 positive and negative.

has been previously reported and attributed to the importance of the conformational epitopes (25).

## Age, gender, and race associations

We expected that the antibody responses collectively assayed within this serosurvey by the microbial antigen MISPA might capture complex relationships between individual antibodies and both demographic and technical variables. We therefore performed a variance partition analysis (26) which models the proportions of variability in antibody responses that are explained by known covariates within these data (Fig. S5). Overall, we observed the largest driver of variability in the data was "participant ID," indicating high relative stability and self-similarity between repeat measures for an individual via the microbial antigen MISPA assay across the time points represented within this study. We also observed antibody responses driven by other covariates, including study variables such as "time point," as well as demographic covariates: "age group," "race," and "gender." Interestingly, our analysis showed a large proportion of unexplained variability ("residuals") in the collective antibody response, indicating the potential importance of additional unmeasured variables or even stochastic influence on an individual's antibody profile. Future studies should elucidate these further.

We further explored systematic differences in antibody response among participants stratified by age group (Fig. 6) while controlling for other covariates. Given that participant ID was such a larger driver of variability, adjusting for participant ID in our analysis was not feasible, despite the comparatively large sample size. We therefore opted to subset data for this analysis to a single sample per individual subject, selecting the most recent sample for any individual who had participated in both time points.

We observed an age-dependent increase in SINR values for Epstein-Barr virus (EBV) BFRF3 antibodies (Fig. 6A), consistent with previous reports (27). Notably, the overall prevalence of responses to EBV was 79.1%, which is less than in other studies (19), which may be due to the young average age of this population, 28.5 ± 12.3 years old (Table S4). We also observed an age-dependent variability in antibodies suggestive of immunization schedules. For example, we observed significantly higher antibody levels against rubella M33 virus among subjects >60 years old (Fig. 6B). This may indicate a residual adaptive response to rubella infections that occurred prior to the introduction

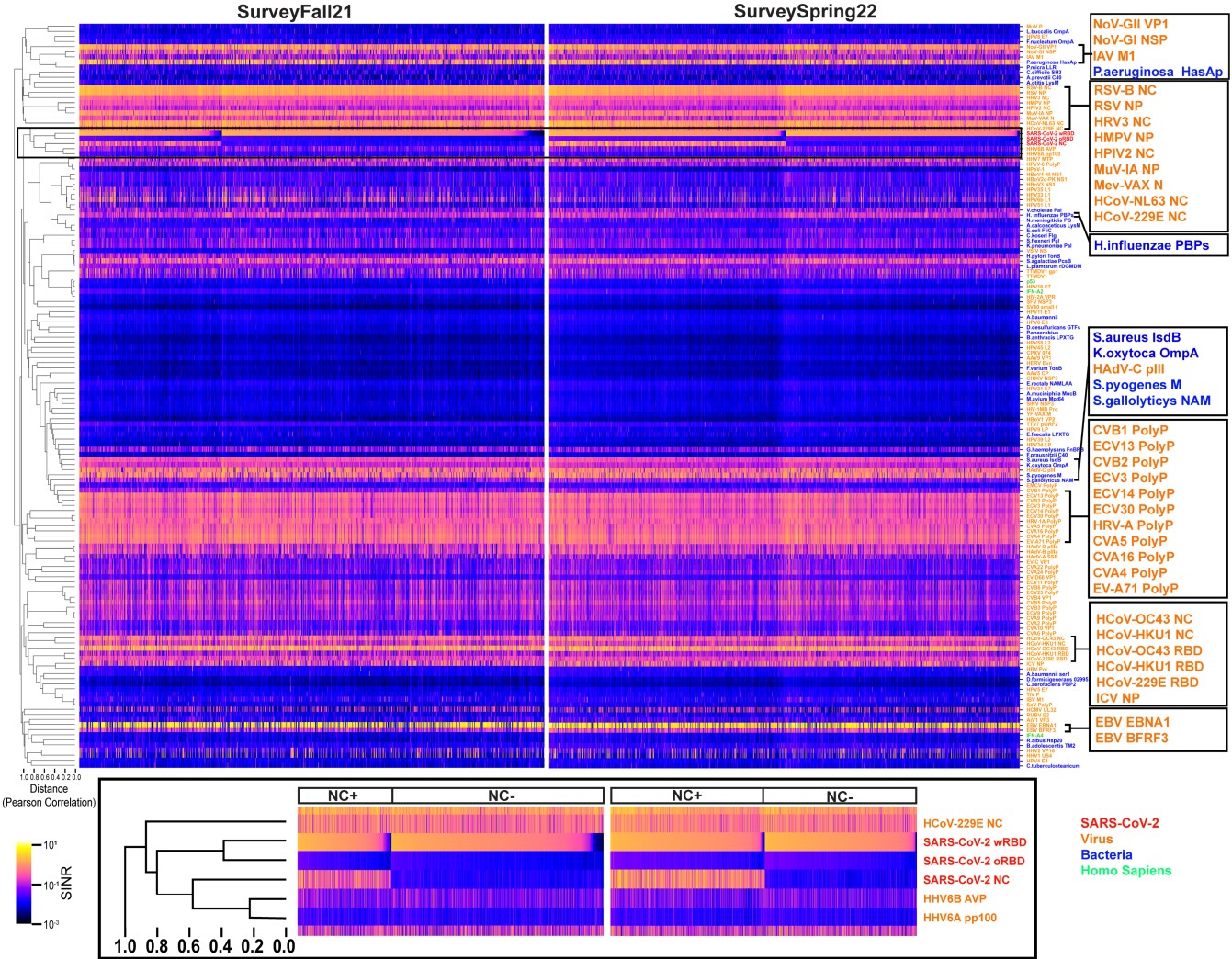

**FIG 4** Seroreactivity heatmaps for 147 antigens from SurveyFall21 (left) and SurveySpring22 (right). Samples are in columns; antigens are in rows; and clustering is based on Pearson correlation coefficient value. The distance labeled at the left-bottom corner was calculated as "1-R." The antigen order is the same in both heatmaps. The antigens of SARS-CoV-2, virus, bacteria, and *Homo sapiens* are color coded in red, orange, blue, and green, respectively. The SARS-CoV-2 antigen responses are highlighted in black rectangle. Antigens with high responses are enlarged on the right panels. SINR, spike-in normalized ratio.

of the rubella vaccine (measles, mumps, and rubella) to the USA in 1969, spurred by the rubella pandemic during 1964–1965 (28).

We also observed significantly higher antibody levels in subjects <30 years old against several HPV antigens that are present within HPV vaccines administered within the USA. Figure. 6Ci shows an increased abundance of antibodies against HPV6b Major Capsid L1, and even more strongly among participants <20 years old. When we stratified these young adult age groups according to self-reported gender, we observed that this finding was largely driven by female subjects (Fig. 6Cii), consistent with previous reports of higher HPV vaccine adoption among females (29).

We also identified antibody levels that vary as a function of participants' self-reported race, including human cytomegalovirus (HCMV) UL32 (Fig. 6D), which revealed an unusually high response among participants who identified as "Asian." This trend was observed in both of our serosurveys, which were largely non-overlapping cohorts. HCMV is more frequently found outside the USA, including Asia (30), and it is notable that among the ASU community, self-reported Asians are the most likely group to include

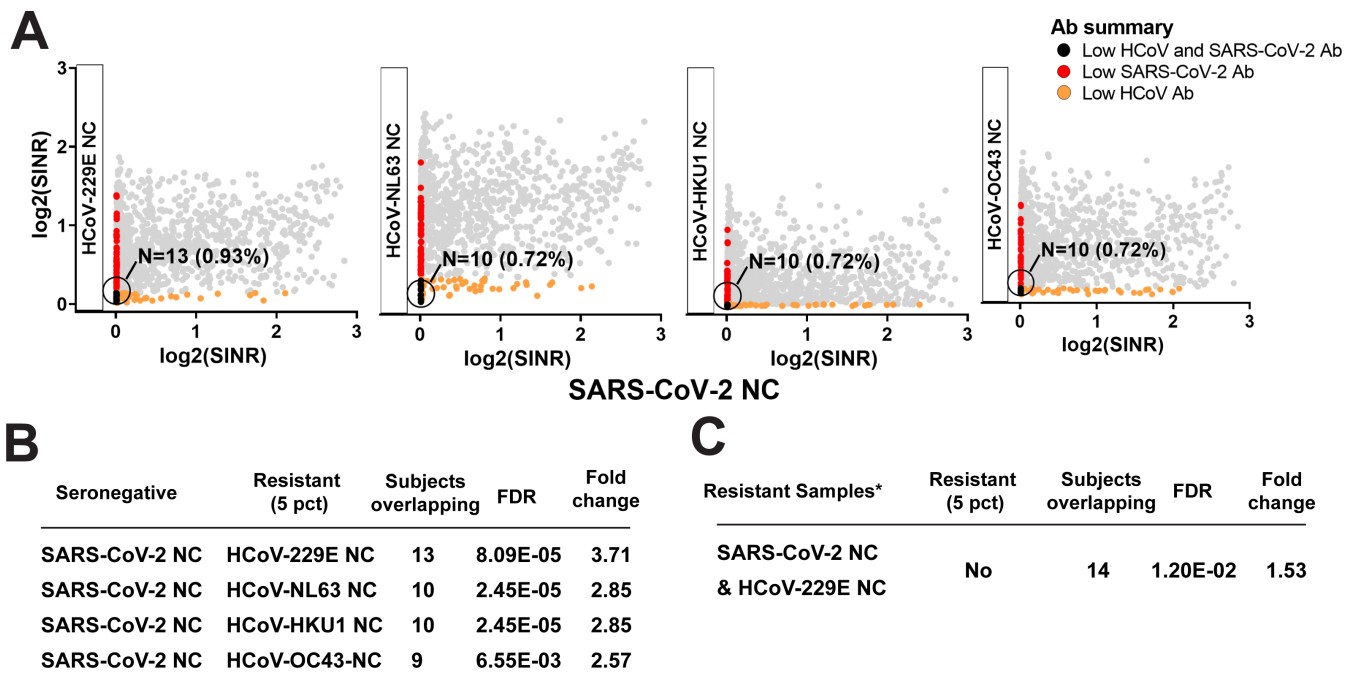

**FIG 5** Serology responses of coronavirus-resistant subpopulations. Low antibody response to sCoV is associated with reduced rates of self-reported COVID-19. (A) Scatter plots between antibody abundance against SARS-CoV-2 and sCoV NC. (B) Overlap between subjects within the lowest fifth percentile of antibody levels indicates an unexpectedly large subpopulation of participants with low antibody response against both sCoV (Resistant Set 2) and SARS-CoV-2 NC (Resistant Set 1) (one-sided Fisher's exact test). (C) Subjects with the lowest fifth percentile for both SARS-CoV-2 and human coronavirus (HCoV)-229E NC antibody levels were at increased likelihood for reporting to have never had COVID-19 (one-sided Fisher's exact test). N denotes the number of subjects; X% denotes N as the percentage of the study population. SINR, spike-in normalized ratio.

visiting scholars. Alternatively, there may be increased prevalence of HCMV among American Asians. Additional studies will be needed to better elucidate this observation.

## Consistency of antibody levels over time

There were 137 subjects who participated in both serosurveys, providing more than 20,000 pairwise comparisons of antibody responses over a 6-month interval. Although participants revealed a broad range of responses to the same antigen, we were struck by the stability of the quantitative antibody responses to nearly all antigens within each participant during this time (Fig. 7A through D; Fig. S7). Notably, these responses were so stable that any deviation suggested an intervening clinical event. The most common of these deviations was a greater than five fold increased anti-NC response observed for 51 participants (e.g., Fig. 7B and D; Fig. S7), the majority of whom [41 (80.4%)] self-reported having COVID-19 sometime between the two surveys in their intake forms (Fig. 7E). Conversely, among participants who did not report having COVID-19 between the two surveys, only 11.8% revealed a fivefold change in anti-NC response, presumably representing asymptomatic infections. Other examples shown here include a participant with >5-fold increase to antigens found in the HPV vaccine, and another with >5-fold increased response to an antigen in the flu vaccine, both of whom reported having received the respective vaccines during this period (Fig. 7C and D).

It was also useful to compare longitudinal trends for participants to the overall population responses (Fig. 7F; Fig. S6). Participant SNP126176 had a response to SARS-CoV-2 NC in SurveyFall21 that was already higher than the 75th percentile of the population, suggesting a prior SARS-CoV-2 infection. Moreover, this response increased further in SurveySpring22, implying another infection. Indeed, this participant reported having COVID-19 twice, once in December 2020 and again in February 2022. This individual's response to influenza A virus protein M1 also showed an interval increase

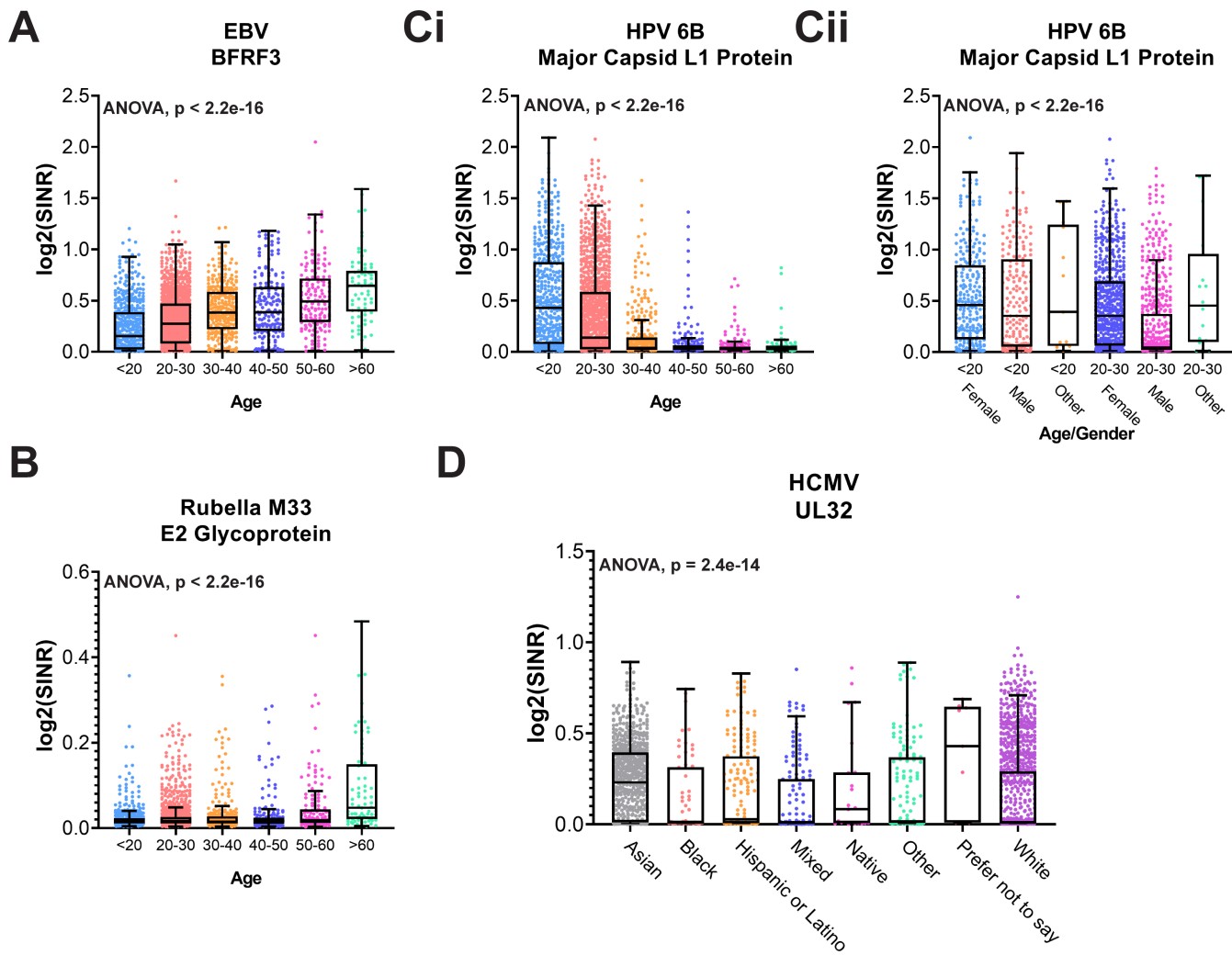

**FIG 6** Selected antibodies associated with age, gender, and race. Boxplots showing age-associated increase in antibody levels against (A) EBV capsid antigen BFRF3 and (B) rubella E2 glycoprotein. Increased antibody levels against HPV 6b capsid antigen L1 in young adult participants (Ci) were driven predominantly by female study participants (Cii). (D). Variability in antibodies against human cytomegalovirus as a function of self-reported race. Group differences are estimated using one-way analysis of variance (ANOVA), adjusting for gender, race, COVID-19 history, COVID-19 vaccination history, SARS-CoV-2 qPCR, and time point. Unadjusted *P* values are shown. SINR, spike-in normalized ratio. Boxes indicate the median and 25th–75th percentile, and whiskers max and min samples.

that corresponded to the self-report of receiving flu vaccine between the two survey dates.

We specifically looked at differences in response to the SARS-CoV-2 antigens for all the 137 participants. As expected, all three, the SARS-CoV-2 wRBD, oRBD, and NC showed significant increases (Fig. S8) using the rank-Sum test, consistent with known new cases during that time window. The SARS-CoV-2 oRBD showed the highest significance (*P* < 0.0001), which agreed with the Omicron wave in early 2022, thought notably the signal strength for that protein is weaker than that for the wRBD. Around half of the population had anti- SARS-CoV-2 NC increased as observed in Fig. 7E.

## DISCUSSION

The current configuration of MISPA supports the analysis of up to 200 different antigens in up to 2,000 samples for a single run. Adding antigens in the same assay has demonstrated neither cross-inhibition nor loss of sensitivity. Notably, the LOD for the same commercial wRBD antibody for SARS-CoV-2 did not change whether measured in an assay with two other proteins or 146 other proteins. However, the addition of more

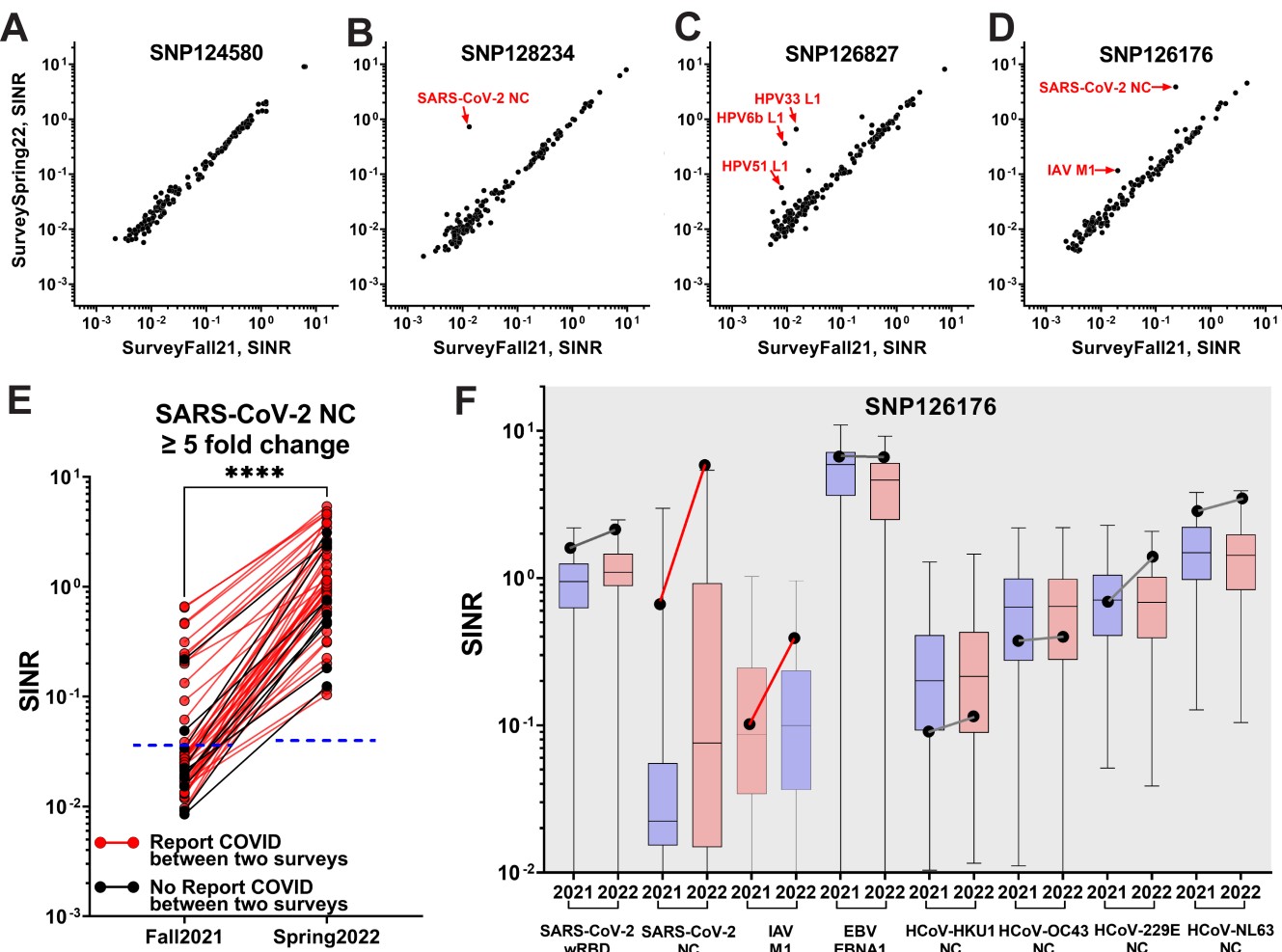

**FIG 7** Longitudinal comparison of seroreactivity against 147 antigens between the two serosurveys (A–F). The SINR of 147 antibodies were plotted as scatter plots. Antigen labeled for ≥5 fold change. (A) Participant SNP124580 had no changes. (B) Participant SNP128234 reported a SARS-CoV-2 infection between the two surveys. (C) Participant SNP126827 reported receiving an HPV vaccination between the two surveys. (D) Participant SNP126176 reported COVID-19 infections in December 2020 and February 2022 and receiving influenza vaccine between the two surveys. (E) Individuals with ≥5-fold increase in anti-SARS-CoV-2 NC response. Participants who reported having or not having COVID-19 infection between the two surveys are indicated in red or black, respectively. The cutoffs for anti-NC seropositivity are indicated as blue dotted lines based on the contrived pre-2019 pooled serum (mean + 3× SD). The anti-NC response between Fall21 and Spring22 was significantly different (two-tailed paired *t*-test).(F) Levels of eight antibodies for Participant 126176 (black dots) at two different time points (2021, SurveyFall21; 2022, SurveySpring22) compared to the distribution of levels for 137 participants (box indicates median and 25th–75th percentile, whiskers max and min samples). All 147 antibodies are shown in Fig. S6. Changes in the participant's antibody levels between the two time points are indicated by red/black lines to indicate ≥or ≤ 5 fold change, respectively. ****, *p* < 0.0001.

samples and antigens requires greater NGS depth to achieve the desired assay dynamic ranges. About ~200 million reads were obtained for our microbial antigen MISPA. A sequencing run with 1 billion reads would readily support assaying 200 antibodies in 10,000 samples, costing 5 cents per antigen per sample.

MISPA builds on the success of other approaches that have assessed immune response detection by NGS (8, 10–12, 31). These methods either use peptides (30–56 amino acids) as target antigens and/or have typically been run on fewer samples (8–12). An advantage of MISPA is the use of full-length protein, allowing the detection of antibodies to conformational epitopes. This may explain why we have observed somewhat more frequent responses to common viruses than reported for the peptide assays. Among highly prevalent viruses such as human respiratory syncytial virus, rhinovirus A, and enteroviruses A and B, MISPA detected nearly 100% responses, compared to less than 75% seroprevalence in a peptide-based assay, albeit these

were measured on different populations (8). However, the peptide-based methods enable epitope mapping, which is not supported in MISPA, making the two approaches complementary.

The MISPA assay addresses an important need for multiplexed antibody testing of many samples. It is currently applied for use in runs with up to 2,000 samples, which could be scaled further with automation and more personnel. The ability to monitor many antigens simultaneously with a microliter of serum makes this test ideal for molecular epidemiology studies, monitoring exposures to many potential pathogens in large populations, even under conditions where clinical sample collection is challenging. Antibody profiles in individuals who were sequentially tested at 6-month intervals were nearly identical, demonstrating that the assay is both reproducible and quantitative for all antigens. The assay is sensitive, with a LOD around 50 ng/mL, irrespective of the number of tested antigens, and has a dynamic range exceeding three log10 units. The LOD for detecting the RBD protein here was less than has been reported by other assays. It is difficult to compare our LOD to other assays without doing a side-by-side comparison. Ultimately each LOD measurement depends on how the experiment was performed, especially the target protein measured. Our measured LOD relied on the monoclonal antibody we used to bind RBD, which has its own affinity for the protein and its own behavior in assays. In these experiments, our focus was to develop an assay that could quantitatively analyze multiple antibodies, and we did not optimize for LOD, which is something that will be needed in the future. The current configuration supports 200 different antigens, which is likely to be sufficient for most clinical scenarios. Nonetheless, the number of antigens could increase with greater sequencing depth, limited primarily by the cost of preparing more proteins. A key advantage of the platform is that there is no requirement for highly specialized equipment, once the MISPA library is produced. Sample incubation, antibody pull-down, barcode amplification, and sequencing readout can all be performed using commonly found lab equipment.

The constant level of antibody responses over a 6-month time window is much longer than the half-life of antibodies [<4 weeks (32)]. This quantitative stability against so many antigens agrees with the known durability of vaccine responses (33) and implies some form of feedback regulation that maintains specific antibodies at a fixed level in the blood over time. This regulation may occur at the level of plasma cells, which are known to survive for longer than a year or may include a more complex mechanism including memory B cells (33–37). Understanding the factors responsible for setting antibody levels, as well as the mechanisms for maintaining them, will contribute to improved disease tracking and offer the potential for immune adjustment.

Multiplexed tools like MISPA that can be applied to populations will enable broader questions regarding how immune profiles predict general health, how they change over time, as well as how they affect responses to new or existing pathogens. However, a remaining challenge for MISPA is determining which pathogen proteins are the best to use to track infections, as some are likely to be more immunogenic or more clinically relevant than others. For example, there was a notable difference in seroprevalence for antibodies against the two mumps antigens included in our library (Table S4). For most antigens, other than SARS-CoV-2, we did not have documented clinical positive and negative samples that would enable us to assess the sensitivity and specificity of the antigens as a measure of infection; hence, the interpretation of the seropositivity was based on the analytical cutoff as described in Materials and Methods. For antigens, except SARS-CoV-2, their serological performance and the antigenic of the protein would need further evaluation individually to understand their reactivity with available true positive and negative samples. There were 30 proteins for which we did not observe a response (≤5.0% seroprevalence) in this study, which featured a largely healthy population. However, 20 of these proteins showed responses in other studies we have done, particularly studies that included cancer or AIDS patients, suggesting that they are antigenic. There were 10 antigens that have showed low response in all of our studies to date, including antigens from hepatitis B virus and yellow fever

virus. For these 10 antigens, the low response could be because no one in these populations had responses to these organisms or because the antigens in the assay are not good probes for measuring responses to those organisms. We hope to develop each antigen by evaluating samples from known positive cases and known negative controls to determine the sensitivity and specificity of each antigen. A combination of bioinformatics, discovery serological studies, and biomedical judgment will aid these decisions.

In addition, MISPA requires defined triage methods to determine the preferred method for producing each protein. Cell-based production is more cumbersome but necessary for some proteins. The majority of proteins were produced in MISPA using an IVTT coupled expression system, which has been employed extensively in the production of tens of thousands of proteins for use in serological studies (13–19). This HeLa cell lysate-based IVTT, which includes both human ribosomes and chaperone proteins, can produce folded and enzymatically active proteins (20). As extracellular domains and proteins reside in a different biochemical environment, the IVTT does not always yield well-folded extracellular proteins, perhaps due to the lack of disulfide bonds and other post-translational modifications. Thus, we produced RBD proteins using Expi293F cells when the IVTT-produced protein did not appear to be immunogenic.

The current MISPA assay was based on the profiling of IgG antibodies through protein G-IgG capture. It would be straightforward to also evaluate IgA and IgM antibodies, which can be performed with the appropriate capture system to profile their response along with IgG, to empower the understanding of the immune response in a further depth. It would be interesting to also understand the correlation between the SARS-CoV-2 RBD-binding Abs and the SARS-CoV-2 neutralization titer. The implementation of a neutralization antibody assay based on competition with their binding with specific antigen would be another possible use of MISPA.

Finally, despite the $10^3$-fold dynamic range in a single-assay run, some antibody responses span an even larger range. Thus, accurate quantification may require running the same sample at more than one serum dilution. Fortunately, this can be accomplished by running the same sample at different dilutions in the same run.

The strength and character of the immune response to target antigens depend on a confluence of factors including genetics, immunoglobulin locus rearrangement, tolerance, and antigenic exposure. Evidence here suggests that it achieves an apparent steady state that is remarkably stable over time. This stability will prove useful for tracking new trends in health conditions as they emerge. The ability to evaluate quantitatively a broader range of immune responses in many samples will provide rich exposure and immunity information in populations and individuals.

## MATERIALS AND METHODS

### Participant population and sample collection

A total of 1,064 and 1,397 participants at least 18 years of age were recruited from the ASU community without severe symptoms of COVID-19 on 13–17 September 2021 (SurveyFall21) and on 1–3 March 2022 (SurveySpring22). Participants included both students and employees. Females and males 18–80 years of age who were able to provide informed consent to provide their data and samples for research were included. Anyone with current upper respiratory symptoms, unable to provide consent, or pregnant women were excluded. Blood samples were collected from each participant.

Participants were asked to complete a baseline questionnaire including age, gender, race, and status (student or employee) and general health information related to COVID-19 (previous COVID-19 test results, symptoms if previously infected, and vaccination dates).

Blood samples were collected from all participants by phlebotomists using BD Vacutainer Venous Blood Collection serum separation tubes (SSTs). SSTs were allowed to clot at room temperature (RT) for 1 hour and stored at 4°C for up to 8 hours before

centrifugation at 1,300 × *g* for 20 min at room temperature. The top serum layer was carefully collected, aliquoted, and stored at −80°C before use.

To assess SARS-CoV-2 MISPA assay's efficacy, there were 64 pre-2019 samples used for evaluation of the efficacy of SARS-CoV-2 analysis that had been collected through a previous unrelated study under the institutional review board of STUDY00009580. Colleagues at the Mayo Clinic Clinical Testing Laboratory kindly provided 55 qPCR test-confirmed COVID-19 deidentified samples collected between 0 and 7 days, 8 and 14 days, and >14 days after symptom onset in early 2020.

## Commercial assay for RBD and NC antibodies

RBD and NC antibodies in each serum sample were tested using assays with FDA EUA in the Clinical Laboratory Improvement Amendments (CLIA) certified Arizona Biodesign Clinical Testing Lab according to the manufacturers' instructions using Access SARS-CoV-2 IgG Antibody Test on a Beckman UniCel DxI 800 instrument (Beckman, Indiana) and the Platelia SARS-CoV-2 total Ab ELISA assay from Bio-Rad, respectively.

Access SARS-CoV-2 chemiluminescent IgG II assay from Beckman Coulter is two-step enzyme immunoassays (Beckman, DXII RBD IgG). These assays were run on a dedicated Beckman UniCel DxI 800 instrument. Microtiter plates containing 200 µL of serum per well were loaded and processed according to the manufacturer's instructions. The result was compared to the cutoff value defined during the calibration of the instrument. The result was interpreted based on the manufacturer's recommendations: <10 AU/mL, negative; ≥10 AU/mL, positive.

Platelia SARS-CoV-2 total Ab ELISA assay from Bio-Rad is a qualitative diagnostic test (Bio-Rad NC ELISA). It is the detection of total antibodies (IgM/IgA/IgG) against SARS-CoV-2 nucleocapsid using fully automated EVOLIS system (Bio-Rad). Specimen results are calculated using the signal to cutoff control value. The specimen result was expressed as a ratio and interpreted based on the manufacturer's recommendations: ≤0.8, negative; between >0.8 and <1.0, equivocal; ≥1.0, positive.

## MISPA library generation

### Cell-based protein expression

RBDs for SARS-CoV-2 and the seasonal coronaviruses were expressed in the Expi293F human expression system (Gibco, Maryland, USA) as secreted proteins. In brief, the gene encoding each RBD with the signal peptide of human tissue plasminogen activator and expression tags in the following configuration: (MDAMKRGLCCVLLLCGAVFVSP)-RBD protein-(GGGGS)$_3$-HaloTag-3xFLAG, was synthesized into the pcDNA3.4 expression vector. Plasmid DNA was prepped with NucleoBond Xtra Midi kit for transfection-grade plasmid DNA (Macherey-Nagel, PA, USA). Expi293F cells (Gibco) were grown in Opti-MEM Reduced Serum Medium (Gibco) and transfected using the ExpiFectamine293 Transfection Kit (Gibco). The RBD proteins were expressed in Erlenmeyer Flasks (VWR International, Pennsylvania, USA) at 37°C with 8% $CO_2$ on an orbital shaker (Laboratory Supply Network, USA). At 4.5 days post-transfection, the cells were harvested and centrifuged 20 min at 4,000 × *g* at 4°C. The supernatant was collected at 3,000 × *g* for 25 min at 4°C, filtered using a 0.22-µm Stericup filter, and stored at −80°C until usage.

### Cell-free protein expression

Genes of interest were cloned into pJFT7_cHalo-3xFLAG destination vector (DNASU, Arizona, USA), and proteins were expressed using a cell-free IVTT (Thermo Fisher, USA). For each 1 mL of IVTT expression mixture, 500-µL HeLa lysate, 200 µL of accessory proteins, and 400 µL of reaction mix were mixed before adding 200 µL of 250-ng/µL plasmid DNA for each protein. The IVTT mixture was incubated at 30°C for 2 hours to allow the production of target proteins with a C-terminal fusion HaloTag and 3xFLAG.

## Protein expression quality control

Functional and full-length protein expression was verified by an in-gel fluorescence assay. Two microliters of expression lysates was mixed with 8 µL of 0.75-µM HaloTag Alexa Fluor 660 Ligand (Promega, Wisconsin, USA) diluted in nuclease-free water, incubated at 25℃ for 30 min under dark. The mixture was mixed with 2.5 µL of XT gel loading dye (Bio-Rad), boiled at 95℃ for 10 min, and loaded in 4%–20% SDS-PAGE gel. The fluorescence signal was measured using a laser-scanner platform (Typhoon, Cytiva, USA).

## Chloroalkane-modified oligo synthesis

To make chloroalkane-modified oligo, 350 µL of 10-mM azido chloroalkane (Acme Bioscience, California, USA) [1-azido-18-chloro-3,6,9,12-tetraoxaoctadecane ($C_{14}H_{28}ClN_3O_4$); this product is customized and synthesized similar to the product from Iris Biotech, Halo-PEF (4)-azide (RI-3710)], 1,000 µL of 1-mM octadiynyl dU modified oligo U1 [Universal Oligo 1, AAAAAAAAAAAAAAAATAGGCCGTTGACTCATCTACG; Integrated DNA Technologies (IDT), New Jersey, USA], and 600 µL of the mixture of 166.7-mM Tris(3-hydroxypropyltriazolylmethyl)amine and 33.3-mM CuSO4 and 700 µL of 0.5-M sodium L-ascorbate solution were mixed and incubated at 37℃ for 16 hours with gentle stirring. The resultant solution was mixed with 265 µL of 3-M sodium acetate and 13,250 µL of pre-chilled 100% ethanol to precipitate the DNA, incubated at −80℃ for 1 h, and centrifuged at 13,000 rpm for 30 min at 4℃. The pellet was washed twice with 5.0-mL pre-chilled 80% ethanol and resuspended into 5-mL nuclease-free water. The concentration of the chloroalkane-modified oligo was qualified using 15% Criterion TBE-Urea Polyacrylamide Gel (Bio-Rad) with serially diluted octadiynyl dU modified oligo U1 as a standard curve.

## DNA barcode production

Ten microliters of 10-µM DNA oligo with a 12-bp unique barcode that has a complementary oligo 1 (CU1) at the 5′ end and complementary oligo 2 (CU2) at the 3′ end (IDT) was annealed with 10 µL of 10-µM chloroalkane-linked oligo at 1:1 ratio by heating to 95℃ for 5 min and gradually cooled down to 25℃ over the course of an hour. The resulting product (20 µL) was further mixed with 5 µL of 20× deoxynucleotide solution mix (NEB, Massachusetts, USA), 10 µL of 10× NEBuffer 2 (NEB), 0.5 µL of 5,000-U/mL Klenow fragment (3′→5′ exo−) (NEB), and 64.5 µL of nuclease-free water, and incubated at 37℃ for 30 min to fill in the fragment for a chloroalkane-linked dsDNA barcode.

## Protein barcoding

Twenty microliters of the expressed protein and 20 µL of 10-µM unique chloroalkane-linked barcode were added to one well in a 96-well plate and incubated at room temperature (25℃) to allow the covalent bond between the chloroalkane and HaloTag to form. The protein expression level from IVTT used in this article could express up to 100 µg/mL based on the vendor, which was up to 1–2 µM for an average size of 50- to 100-kDa protein. The concentration of the barcode was at least five times higher than the target protein to allow the full barcoding. The mixture was then transferred to another 96-well 1,000-µL deep well plate containing 100 µL of anti-FLAG coated magnetic beads (2.5% slurry) (A36797; Thermo Scientific, Massachusetts, USA). Finally, the 1,000-µL deep well plate was incubated at 4℃ on an orbit shaker and shaken at 800 rpm for 16 h. After the anti-FLAG beads captured the barcoded protein through the 3xFLAG, the proteins were eluted with 100 µL 0.5-mg/mL 3xFLAG peptide (Thermo Scientific) solution in 1× Tris-buffered saline (TBS) with 20% glycerol after two times of 1× TBS with 0.2% Tween 80 and two times of 1× TBS wash.

## Protein barcoding quantification

Barcoded protein (2.5 µL) diluted 1:100 in nuclease-free water was added to a PCR mixture containing 5 µL of 1-µM forward, 5 µL of 1-µM reverse primer, 0.5 µL of 10-pM spike-in control oligo [synthesized dsDNA with 12-bp unique DNA barcode (GTGAAGCT TACG)], and 12.5 µL of 2× sapphire PCR master mix (Takara Bio, USA). The sample was mixed well and run in PCR, 1 cycle of 94°C for 1 min; 16 cycles of 98°C for 15 seconds, 60°C for 10 seconds, and 72°C for 10 seconds; and 1 cycle of 72°C for 15 seconds. There were 147 proteins tested in the current study. The mixture of protein barcodes were further barcoded with a sample-specific barcode by adding unique indexes through PCR. Then, 10 µL of PCR product from each sample was pooled. The pooled PCR products from all the samples were purified using a QIAquick PCR purification kit (Qiagen, USA) following the manufacturer's instructions. The purified PCR product was quantified by qPCR, denatured by 0.2-M NaOH, and diluted to 20 pM using hybridization buffer HT1 solution (Illumina, USA). A mixture of 140 µL of 20-pM denatured PCR product, 130 µL of 1.8-pM denatured PhiX (Illumina), and 1,030 µL of HT1 solution was made for Illumina NextSeq550Dx analysis. The NGS counts for individual barcodes were divided by the corresponding counts of the spike-in control oligo as an SINR, and a barcoded protein SINR value higher than the negative controls that did not have expressed protein (nuclease-free water, mean + 3× SD) was considered successful protein barcoding.

## MISPA library generation and quantification

To make the three-antigen MISPA library that contained SARS-CoV-2 wRBD, NC, and the GFP, the final SINR value for each barcoded protein was diluted according to the quantification analysis. For the 147-microbial antigen MISPA library, each barcoded protein that passed barcoding quantification was mixed as an equal volume. As a quality control measure and to establish a baseline assessment of species abundance for each barcoded protein, we analyzed each MISPA library by amplifying the protein barcodes of the entire library with 16 cycles of PCR and then analyzing the products by NGS analysis. Absence of a protein (or linked barcode) was defined as sequence read counts lower than the mean plus three times the standard deviation of the negative controls [1% bovine serum albumin (BSA)].

## MISPA serology assay

### Protein capture

The MISPA serology assay started with production of a MISPA library. From the MISPA library, 20 µL was aliquoted into each well of a 300-µL 96-well plate for use. Serum/plasma containing specific antibodies was diluted 1:5 using buffer containing 1× TBS and 20% glycerol. Five microliters of diluted sample was then added to an individual well containing a 20-µL MISPA library. The plate with the mixture was mixed briefly using a benchtop vortex for 10 seconds at the maximum speed and then incubated on an orbit shaker, shaking at 800 rpm for 1 h at RT (25°C). The mixture was then transferred to another 96-well 1,000-µL deep well plate containing 100 µL of protein G-coated magnetic beads (contains 20% slurry). The 1,000-µL deep well plate was shaken on an orbit shaker at 800 rpm for 16 h at 25°C. After antibody-antigen complex capture by the protein G beads, the beads were washed on a KingFisher Flex System (Thermo Fisher) configured for 96-well processing, twice with 500-µL 1× TBS + 0.2% Tween 80 and twice with 500-µL 1× TBS. After washing, the protein G beads were resuspended in another 100 µL of 1× TBS buffer.

### Barcode collection

To each well of a 300 µL PCR 96-well plate was added: 10 µL of sample-specific index pairs (unique forward and reverse primers), 0.5 µL of 10 pM Spike-in control oligo, and 12.5 µL of 2 × sapphire PCR master mix. Then, 5 µL of the antibody-antigen complex protein G bead slurry (20%) in 1x TBS buffer were transferred to the PCR plates for PCR

amplification. The mixture was mixed with a benchtop vortex and run in PCR, 1 initial cycle of 94°C for 1 min; 16 cycles of 98°C for 15 seconds, 60°C for 10 seconds, and 72°C for 10 seconds; and 1 final cycle of 72°C for 15 seconds. The resulting amplified fragments, which contain both protein-specific and sample-specific barcodes, were mixed into a pool by adding 10 µL of each PCR product. The pooled product was purified using a QIAquick PCR purification kit (Qiagen) following the manufacturer's instructions.

### NGS analysis

The purified PCR product was quantified by qPCR, denatured, and diluted to 20 pM using a hybridization buffer HT1 solution (Illumina). A mixture of 140 µL of 20-pM denatured PCR product, 130 µL of 1.8-pM denatured PhiX (Illumina), and 1,030 µL of HT1 solution was made for Illumina NextSeq550Dx analysis. The NGS counts for individual barcodes were divided by the corresponding counts of the spike-in control oligo to produce the SINR.

The SurveyFall21 and SurveySpring22 serosurvey were analyzed in two independent MISPA assays separately using the same protein library. For each MISPA assay, we included three controls, positive control (contrived SARS-CoV-2 positive sample pool, SARS-CoV-2 wRBD positive and NC positive), negative control (contrived pre-2019 serum sample, both SARS-CoV-2 wRBD and NC negative, 16 replicates), and blank control (1% BSA diluted in 1× TBS with 20% glycerol). The negative control was used to generate the seropositive assay cutoff for SARS-CoV-2 wRBD and NC (mean + 3× SD), and the blank control was used to generate the technical cutoff (5× mean) for seroresponse of all antigens. The MISPA library used in each MISPA assay was diluted by 1:100 in 1× TBS and tested as a mock pull-down sample, and the result was used to verify the presentation of each barcoded protein. A total of 185 samples were randomly selected from SurveySpring22 that were repeated in SurveyFall21 MISPA analysis to access the reproducibility.

### MISPA NGS data processing

Sample-deconvoluted FASTQ files from Illumina NextSeq550Dx were quality-checked for total reads (>200 M) and sequencing quality (mean Phred score >28), and, for each sample, the sequencing reads were then mapped to proteins by matching the first 12 bp of sequences to the protein barcodes by using a custom Python script. The SurveyFall21 and SurveySpring22 serosurveys had 225- and 297-M total protein-mapped reads (or around 135,000 and 170,000 mapped reads per sample), respectively. The barcode count for each protein was then normalized to the SINR by dividing by the barcode count of spike-in oligo in each sample/assay.

### Statistical analysis

Reproducibility of response against all antigens for 185 samples, the commercial pre-2019 contrived serum sample (Fig. 3D and E), and the response of individual antigens common to PL_147 and PL_184 (Fig. 3F) were assessed through a linear regression model and $R^2$ value. The seropositivity for each antigen was calculated separately for SurveyFall21 and SurveySpring22 based on the technical cutoffs (5× mean) of the blank control. The Pearson correlation coefficient was used as the distance metric for clustering analysis for SurveyFall21, and the same cluster was used to generate a heatmap of SurveySpring22 (Fig. 4). The correlation clustering patterns across all antigens for two surveys was assessed through Rand index analysis (Fig. S3). To understand the serology responses of coronavirus-resistant subpopulations, the fifth percentile of abundance for each of the seasonal coronaviruses was analyzed by the one-sided Fisher's exact test (Fig. 5). The antibody differences between self-reported COVID-19 positive and negative (confirmed with Bio-Rad Platelia SARS-CoV-2 total Ab ELISA assay anti-NC negative) were compared by Wilcoxon rank-sum test. Antibody responses across age, gender, and race groups were analyzed using one-way analysis of variance (Fig. 6). To evaluate antibody

changes over time, we further analyzed the 137 subjects who participated in both surveys, and a five fold change in either direction was taken as an arbitrary cutoff for detecting antibody level changes (Fig. 7; Fig. S7). Overall anti-SARS-CoV-2 NC responses between he two surveys for 137 common samples were compared by a two-tailed paired *t*-test (Fig. 7E). The response differences of the SR+ and SR− groups in each survey were compared by Wilcoxon rank-sum test (Fig. S4).

## Limit of detection

The LOD of wRBD on MISPA was assessed by analyzing a serially diluted monoclonal mouse anti-RBD antibody (0.1–100,000 ng/mL, 10×) from BD (catalog #MAB105802). We used a five-parameter logistic regression model to estimate the LOD at $y = B + (T + B) / ([1 + 10\text{\textasciicircum}b(\text{xmid-}x)]\text{\textasciicircum}s)$ and its 95% CI. In the formula, $B$ and $T$ are the bottom and top asymptotes when the concentration values go to $-\infty$ and $+\infty$. The values $b$, xmid, and $s$ are the steepness of the curve, $x$-coordinate at the inflection point (where the curve changes direction), and an asymmetric coefficient. The model was also used to find the concentration value corresponding to the LOD value.

## Comparing MISPA to clinical results

To evaluate the performance of MISPA compared to qPCR-verified COVID-19 positive and pre-2019 samples, we calculated the PPA and NPA. The seropositive cutoffs for SARS-CoV-2 wRBD and NC for MISPA were calculated based on the contrived pre-2019 sample (mean + 3× SD). The comparison of MISPA and qPCR-verified COVID-19 positivity was calculated based on the following formulas:

- PPA = (# both assays positive) / (# clinical assay positive) × 100.

- NPA = (# both assays negative) / (# clinical assay negative) × 100.

## Variance partition analysis

Variance partition analysis was performed using the VariancePartition library available in the R programming language. Pan-147 MISPA abundance values were offset by 1 and log2 transformed and offset by 1, and a model was fit for each antigen using the "fitExtractVarPartModel" function, including the following terms: "participant ID," "race," "city," "survey date," "age group," "gender," "time point," "COVID-19," "vaccine," "qPCR result," "status," and interaction terms for "COVID-19: age group" and "COVID-19: gender."

## ACKNOWLEDGMENTS

We thank all the participants in serosurvey studies at ASU and ASU Biodesign Clinical Test Laboratory for sample and metadata collection. We thank Jonathan Blum, M.D., Ph.D., for discussions and suggestions.

This research was supported by grants from Innovative Molecular Analysis Technologies (R21CA196442), Arizona Board of Regents (2950007-01). Specimen collection and analysis for both SurveyFall21 and SurveySpring22 were funded by ASU Knowledge Enterprise. This project has been funded in part with Federal funds from the National Cancer Institute, National Institutes of Health, under Contract No. 75N91019D00024, Task Order No. 75N91021F00001. The content of this publication does not necessarily reflect the views or policies of the Department of Health and Human Services, nor does mention of trade names, commercial products or organizations imply endorsement by the U.S. Government.

Experiment design and data collection: L.S.S., F.R., V.M., J.G.P., J.Q., M.P., H.L., D.A., S.G.R., K.S.A., and J.L.; serosurvey sample and metadata collection: V.M., C.W.H., D.M.M., and G.T.S.; next-generation sequence performance and data analysis: L.S.S., D.A., and J.G.P.; data analysis: L.S.S., J.G.P., B.R., Y.C., J.Q., and J.L.; project supervision and

administration: J.L. and V.M.; writing (original draft): L.S.S., B.R., J.G.P., and J.L.; writing (review and editing): L.S.S., B.R., F.R., V.M., J.G.P., J.Q., Y.C., C.W.H., and D.M.M.

## AUTHOR AFFILIATIONS

[1]Virginia G. Piper Center for Personalized Diagnostics, Biodesign Institute, Arizona State University, Tempe, Arizona, USA

[2]College of Health Solutions, Arizona State University, Tempe, Arizona, USA

[3]School of Life Sciences, Arizona State University, Tempe, Arizona, USA

[4]Arizona State University-Banner Neurodegenerative Disease Research Center, Tempe, Arizona, USA

[5]School of Molecular Sciences, Arizona State University, Tempe, Arizona, USA

## AUTHOR ORCIDs

Lusheng Song  http://orcid.org/0000-0003-2534-5198
Ji Qiu  http://orcid.org/0000-0002-7913-9042
Joshua LaBaer  http://orcid.org/0000-0001-5788-9697

## FUNDING

| Funder | Grant(s) | Author(s) |
| --- | --- | --- |
| HHS \| NIH \| National Cancer Institute (NCI) | R21CA196442 | Femina Rauf |
|  |  | Jin G. Park |
|  |  | Joshua LaBaer |
| Arizona Board of Regents (ABOR) | 2950007-01 | D. Mitchell Magee |
|  |  | Ji Qiu |
|  |  | Vel Murugan |
|  |  | Jin G. Park |
|  |  | Yunro Chung |
|  |  | Joshua LaBaer |
|  |  | Lusheng Song |
|  |  | Femina Rauf |
|  |  | Ching-Wen Hou |
|  |  | Huafang Lai |
|  |  | Deborah Adam |
|  |  | Milene Peterson |
|  |  | Stephen G. Rice |
| HHS \| NIH \| National Cancer Institute (NCI) | 75N91021F00001 | D. Mitchell Magee |
|  |  | Ji Qiu |
|  |  | Guillermo Trivino Soto |
|  |  | Vel Murugan |
|  |  | Jin G. Park |
|  |  | Yunro Chung |
|  |  | Joshua LaBaer |
|  |  | Lusheng Song |
|  |  | Femina Rauf |
|  |  | Ching-Wen Hou |
|  |  | Huafang Lai |
|  |  | Deborah Adam |
|  |  | Milene Petersen |

| Funder | Grant(s) | Author(s) |
|---|---|---|
| | | Stephen G. Rice |
| Arizona State University (ASU) | OKED | Guillermo Trivino Soto |
| | | Vel Murugan |
| | | Yunro Chung |
| | | Ching-Wen Hou |

## AUTHOR CONTRIBUTIONS

Lusheng Song, Conceptualization, Data curation, Formal analysis, Investigation, Methodology, Project administration, Resources, Validation, Visualization, Writing – original draft, Writing – review and editing | Femina Rauf, Conceptualization, Data curation, Formal analysis, Investigation, Methodology, Writing – review and editing | Ching-Wen Hou, Investigation, Methodology, Writing – original draft, Writing – review and editing | Ji Qiu, Conceptualization, Data curation, Formal analysis, Funding acquisition, Investigation, Methodology, Project administration, Supervision, Validation, Writing – original draft, Writing – review and editing | Vel Murugan, Data curation, Formal analysis, Funding acquisition, Investigation, Methodology, Project administration, Resources, Writing – review and editing | Yunro Chung, Formal analysis, Methodology, Resources, Software, Validation, Visualization, Writing – review and editing | Huafang Lai, Data curation, Methodology, Writing – review and editing | Deborah Adam, Data curation, Methodology | D. Mitchell Magee, Data curation, Investigation, Methodology | Guillermo Trivino Soto, Data curation, Methodology | Milene Peterson, Data curation, Methodology | Karen S. Anderson, Data curation, Methodology, Resources | Stephen G. Rice, Methodology, Project administration | Benjamin Readhead, Formal analysis, Methodology, Software, Visualization, Writing – review and editing | Jin G. Park, Conceptualization, Data curation, Formal analysis, Funding acquisition, Investigation, Methodology, Project administration, Writing – original draft, Writing – review and editing | Joshua LaBaer, Conceptualization, Data curation, Formal analysis, Funding acquisition, Investigation, Methodology, Project administration, Resources, Supervision, Writing – original draft, Writing – review and editing

## DATA AVAILABILITY

All data are available in the main text or the supplemental material.

## ETHICS APPROVAL

All volunteer participants in this serosurvey study provided serum and saliva samples under approval by Arizona State University's institutional review board (STUDY00014505).

## ADDITIONAL FILES

The following material is available online.

### Supplemental Material

**Supplemental material (Spectrum02399-S0001.docx).** Fig. S1 to S8; Tables S1 to S4.

### Open Peer Review

**PEER REVIEW HISTORY (review-history.pdf).** An accounting of the reviewer comments and feedback.

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
