## [Reviewer comments · Microbiology Spectrum]

Microbiology Spectrum

Quantitative assessment of multiple pathogen exposure and immune dynamics at scale

Lusheng Song, Femina Rauf, Ching-Wen Hou, Ji Qiu, Vel Murugan, Yunro Chung, Huafang Lai, Deborah Adam, D. Mitchell Magee, Guillermo Trivino Soto, Milene Petersen, Karen S. Anderson, Stephen G. Rice, Benjamin Readhead, Jin G. Park, and Joshua LaBaer

Corresponding Author(s): Joshua LaBaer, Arizona State University Biodesign Institute

Review Timeline:

Submission Date:	June 7, 2023
Editorial Decision:	October 10, 2023
Revision Received:	November 1, 2023
Editorial Decision:	November 2, 2023
Revision Received:	November 2, 2023
Accepted:	November 13, 2023

Editor: Kileen Shier

Reviewer(s): Disclosure of reviewer identity is with reference to reviewer comments included in decision letter(s). The following individuals involved in review of your submission have agreed to reveal their identity: Lakshmanane Premkumar (Reviewer #3)

Transaction Report:

DOI: <https://doi.org/10.1128/spectrum.02399-23>

October 10, 2023

Prof. Joshua LaBaer
Arizona State University Biodesign Institute
Tempe

Re: Spectrum02399-23 (Quantitative assessment of multiple pathogen exposure and immune dynamics at scale)

Dear Prof. Joshua LaBaer:

Thank you for submitting your manuscript to Microbiology Spectrum. I appreciate your patience and submission of additional reviewers due to the unique focus of your manuscript.

Link Not Available

Sincerely,

Kileen Shier

Journals Department
Reviewer comments:

Reviewer #1 (Comments for the Author):

Thank you for the opportunity to review the manuscript "Quantitative Assessment. Of multiple pathogen exposure and immune dynamics at scale". The manuscript describes the use of a multiplexed serology method, MISPA, to evaluate the serological response using a large-scale, high-throughput method with the ability to assess for antigens to 39 bacteria and 99 viruses from a single sample. The method was evaluated in 2400 individuals for two surveillance time periods during the COVID-19 pandemic. The method was found to have high sensitivity, a wide dynamic range and performs favorably compared to commercially available SARS-COV-2 serology assays. In addition, this method captured quantitative longitudinal stability of the serological responses over the time period between the two surveys that could be correlated with new infections or vaccination events highlighting the potential use for the method to be applied as an epidemiology tool. The manuscript presents a comprehensive

overview of the results and methods, making it an intriguing read that would capture the interest of its readers.

Major Concerns:

-Results section: This section could be streamlined to just results. There are some instances of information that might be better suited for the Methods and other instances of discussion of the results (eg. Lines 322-330) that make more sense to be in the discussion section.

-Consider moving Fig. 3 to supplemental and moving the individual demographics Table S2 to the main text. Since there are subanalyses performed (vaccination seropositivity, age, gender etc.) that are addressed in the main text and figures it would be easier for readers to refer to the table in the main article.

-Fig. 9: It is impossible to read any of the antigens that are 5-fold different even with zooming in and therefore, I cannot appreciate how it would be helpful to the reader. I advise revising the figure so that it is legible.

Minor concerns:

-Line 29: specify SARS-CoV-2 "commercial assays"

-Line 84: The authors describe the development of a "clinical testing-compatible" method but do not indicate the turnaround time. I believe readers would be interested in knowing how feasible this method is clinically and therefore describing the time would be beneficial.

-Describe the sources of the pre-2019 and 202 samples in the Methods

-Fig. 7 panels should read "ANOVA"

-Table. 1 should read "Survey"

-Lines 570-611: Statistics are presented in the Results and figures that have not been described in the Methods.

-Line 581: Address how "equivocal" results were handled for performance calculations

Material and Methods: For proprietary names of assay kits and reagents, include the manufacturer followed by location.

Reviewer #3 (Comments for the Author):

This manuscript describes a new platform, called Multiplexed In-Solution Protein Array (MISPA), that can analyze blood samples on a large scale. The MISPA assay utilizes folded protein antigens fused with a halo-tag and coupled to DNA barcodes to evaluate antigen-specific antibodies in serum by NGS. The authors claim that they were able to evaluate antibody responses to 39 bacteria and 99 viruses in 2400 individuals using their MISPA assay. While the reported MISPA assay builds upon other successful methods reported previously for detecting immune responses through NGS, this appears to be the first study to demonstrate the assay's potential for evaluating 100+ antigens (147) and 1000+ clinical samples (2400). Overall, the manuscript is well-written and has the potential to advance immune profiling methods for larger sample sizes. However, I have several concerns that must be addressed before publication.

The authors mentioned using 147 antigens (39 bacterial and 99 viral proteins) to evaluate the MISPA assay. Could the authors provide the details for designing the antigen panel? Additionally, many antigens reported were negative or yielding undetectable signals except certain respiratory viruses and a limited number of pathogenic bacteria. They reported that the samples evaluated were from adults in the ASU community in September 2021 and March 2022, and their seroprevalence estimate matched previous reports. However, it's unclear how the authors confirmed the retained antigenicity of the bacterial and viral antigens upon fusing to the halo tag.

The MISPA assay seems to rely on fusing the antigen with a Halo-tag. Do the authors anticipate that this could pose a problem for the MISPA assay's ability to be more comprehensive, given that many viral glycoproteins may be more difficult to express with Halotag? Additionally, can the authors comment on any concerns about steric hindrance affecting PCR amplification of certain antigens based on their size and shape?

Can the authors explain why the reported LOD of the MISPA assay (~50 ng/ml) is lower than that of traditional ELISA and orders of magnitude less than multiplex assay, for example, Luminex assay platform? Please discuss this limitation in the manuscript.

The authors reported that the PPA for NC using MISPA was 90.9%, and the NPA was 98.4% compared to the EUA-approved test. However, a serosurvey in Table 1 showed that the percentage of people who tested positive for COVID-19 and had detectable antibodies increased from 56.9% in September 2021 to 72% in March 2022 using the EUA-approved test. In contrast, the increase in MISPA assay during this period was from 84.3% to 87.5%. Could the authors explain this discrepancy?

Several studies have reported Ab waning and affinity maturation following SARS-CoV-2 infection or vaccination. However, the authors of this study conducted two serosurveys involving 137 participants and found that quantitative antibody responses to almost all antigens within each participant remained stable during this period. Notably, the authors tested these samples at a serum dilution of 1:5. Can the authors clarify whether the MISPA assay can accurately measure antibody concentration at this serum dilution? How was the 1:5 serum dilution determined? Since the first serosurvey was conducted using samples collected before the emergence of the Omicron variant (in September 2021), and the second serosurvey was conducted during the

Omicron wave (in March 2022), can the authors comment on the difference in antibody reactivity to Wuhan and Omicron antigens, as both these antigens appear to be included in the panel?

Numerous studies have reported the correlation between measured SARS-CoV-2 RBD binding Abs and SARS-CoV-2 neutralization titer. Have the authors tried correlating the RBD binding signal from the MISPA assay to SARS-CoV-2 neutralization titers?

Have the authors tried to identify IgM or IgA antibodies? It is described that the protein G beads were used to pull the immunocomplex, which will be selective for IgG Abs. Please discuss the potential utility of MISPA assay to detect other Ab isotypes.

In lines 164-165, it was stated that the MISPA assay dynamic range for wRBD was from 38.9-100,000 ng/mL of anti-RBD (Fig. 1B). However, data presented in Fig 1B appear to saturate around 10,000. Should the dynamic range be between 38.9 and 10,000?

In line 464, provide the catalog number if Azido chloroalkane is commercially available. Otherwise, please provide details of its synthesis for reproducibility.

In line 487, it was mentioned that 20 µl of the expressed protein was used for halo-tagged protein barcoding. However, considering that expression levels may vary across constructs, giving an estimated range of the protein quantity used in this reaction would be best. How the conjugation efficiency of choloalkane with the halo-tagged protein was evaluated?

In lines 490-491, provide the source and catalog number for anti-flag-coated magnetic beads.

Staff Comments:

Preparing Revision Guidelines

Please return the manuscript within 60 days; if you cannot complete the modification within this time period, please contact me. If you do not wish to modify the manuscript and prefer to submit it to another journal, please notify me of your decision immediately so that the manuscript may be formally withdrawn from consideration by Microbiology Spectrum.

Peer Review Template:

Summary of Key Findings (200-250 words)

Thank you for the opportunity to review the manuscript “Quantitative Assessment. Of multiple pathogen exposure and immune dynamics at scale”. The manuscript describes the use of a multiplexed serology method, MISPA, to evaluate the serological response using a large-scale, high-throughput method with the ability to assess for antigens to 39 bacteria and 99 viruses from a single sample. The method was evaluated in 2400 individuals for two surveillance time periods during the COVID-19 pandemic. The method was found to have high sensitivity, a wide dynamic range and performs favorably compared to commercially available SARS-COV-2 serology assays. In addition, this method captured quantitative longitudinal stability of the serological responses over the time period between the two surveys that could be correlated with new infections or vaccination events highlighting the potential use for the method to be applied as an epidemiology tool. The manuscript presents a comprehensive overview of the results and methods, making it an intriguing read that would capture the interest of its readers.

Major Concerns (at most 5-6):

- Results section: This section could be streamlined to just results. There are some instances of information that might be better suited for the Methods and other instances of discussion of the results (eg. Lines 322-330) that make more sense to be in the discussion section.
- Consider moving Fig. 3 to supplemental and moving the individual demographics Table S2 to the main text. Since there are subanalyses performed (vaccination seropositivity, age, gender etc.) that are addressed in the main text and figures it would be easier for readers to refer to the table in the main article.
- Fig. 9: It is impossible to read any of the antigens that are 5-fold different even with zooming in and therefore, I cannot appreciate how it would be helpful to the reader. I advise revising the figure so that it is legible.

Minor Concerns (at most 5-6 in bullet points):

- Line 29: specify SARS-CoV-2 “commercial assays”
 - Line 84: The authors describe the development of a “clinical testing-compatible” method but do not indicate the turnaround time. I believe readers would be interested in knowing how feasible this method is clinically and therefore describing the time would be beneficial.
 - Describe the sources of the pre-2019 and 202 samples in the Methods
 - Fig. 7 panels should read “ANOVA”
 - Table. 1 should read “Survey”
 - Lines 570-611: Statistics are presented in the Results and figures that have not been described in the Methods.
 - Line 581: Address how “equivocal” results were handled for performance calculations
- Material and Methods: For proprietary names of assay kits and reagents, include the manufacturer followed by location.

-The study was conducted during the COVID-19 pandemic, where the epidemiology of certain respiratory viruses and other pathogens was impacted. Please comment on any bias that this may have introduced into the study.

Confidential Comments to the Editor:

- Although the authors suggest there is a clinical application, I am not sure the discussion supports this. It is however an interesting approach and I believe publishing the methodology is warranted.
- In addition, I was not able to review Figure 9 as it was not legible which I imagine would also be true for the readers.

Song L. T. et al. "Quantitative assessment of multiple pathogen exposure and immune dynamics at scale. Quantitative assessment of multiple pathogen exposure".

In this manuscript, the authors described a multiplexed serology method that evaluates samples at the scale of thousands. It is claimed by the authors that 2400 patients have been evaluated the serological profile to 39 bacteria species/strains and 99 viruses. This screening has been performed by novel multiplexed immunoassays named MISPA.

The current manuscript may be accepted in Microbial Spectrum after addressing all the following comments and aspects to be solved.

Major Concerns

In this work, it is only focused on IgG serological profile; however, it is not discussed and/or studied IgM or IgA as are highly relevant on humoral immune response against pathogens. It is not clear why IgM and IgA are not studied. In addition, it has been previously reported that prior immunity is playing a critical role in immune dynamics and multiple pathogen exposure. The authors should be discussed about prior immunity in the studied cohort and the potential of this MISPA methodology on the analysis of prior immunity. On the other side, conventional protein arrays, as reported by Landeira A. et al. 2023 (Cancer, 2023, 15(3), 891), have demonstrated the ability to decipher multiple pathogen exposure and immune dynamics; in this regard, the authors should consider and discuss these approaches among to provide pros- and cons- about the immune dynamics against main pathogens.

To be accepted all these comments might be addressed.

Response to the Reviewers' comments:

Reviewer #1 (Comments for the Author):

Thank you for the opportunity to review the manuscript "Quantitative Assessment. Of multiple pathogen exposure and immune dynamics at scale". The manuscript describes the use of a multiplexed serology method, MISPA, to evaluate the serological response using a large-scale, high-throughput method with the ability to assess for antigens to 39 bacteria and 99 viruses from a single sample. The method was evaluated in 2400 individuals for two surveillance time periods during the COVID-19 pandemic. The method was found to have high sensitivity, a wide dynamic range and performs favorably compared to commercially available SARS-COV-2 serology assays. In addition, this method captured quantitative longitudinal stability of the serological responses over the time period between the two surveys that could be correlated with new infections or vaccination events highlighting the potential use for the method to be applied as an epidemiology tool. The manuscript presents a comprehensive overview of the results and methods, making it an intriguing read that would capture the interest of its readers.

Major Concerns:

1. Results section: This section could be streamlined to just results. There are some instances of information that might be better suited for the Methods and other instances of discussion of the results (eg. Lines 322-330) that make more sense to be in the discussion section.

Response: We appreciate the reviewer's kind suggestion. We removed some of the information to the methods section or the discussion section to make the results section more streamlined. The changes were marked in blue in the "Marked-Up Manuscript" file.

We removed the "A modified oligonucleotide, 5' 5-Octadiynyl dU modified DNA, was conjugated to the chloroalkane through Cu(I)-catalyzed azide-alkyne cycloaddition" in result section (line 109-110).

We moved the "which has been employed extensively in the production of tens of thousands of proteins for use in serological studies (13-19). This HeLa cell lysate-based IVTT, which includes both human ribosomes and chaperone proteins, can produce folded and enzymatically active proteins (20). As extracellular domains and proteins reside in a different biochemical environment, the IVTT does not always yield well-folded extracellular proteins, perhaps due to the lack of disulfide bonds and other post translational modifications (PTM). Thus, we produced some protein targets using" in the result section to the discussion section (line 411-419).

We moved the "The constant level of antibody responses over a 6-month time window is much longer than the half-life of antibodies (< 4 weeks (38)). This quantitative stability against so many antigens agrees with the known durability of vaccine responses (39) and implies some form of feedback regulation that maintains specific antibodies at a fixed level in the blood over time. This regulation may occur at the level of plasma cells, which are known to survive for longer than a year or may include a more complex mechanism including memory B cells (39-43). Understanding the factors

responsible for setting antibody levels, as well as the mechanisms for maintaining them, will contribute to improved disease tracking and offer the potential for immune adjustment.” to the discussion section (line 377-385).

2. Consider moving Fig. 3 to supplemental and moving the individual demographics Table S2 to the main text. Since there are subanalyses performed (vaccination seropositivity, age, gender etc.) that are addressed in the main text and figures it would be easier for readers to refer to the table in the main article.

Response: We appreciate the reviewer's suggestion. We moved Fig. 3 to figs. S2 and the individual demographics Table S2 to the main text as Table 1.

3. Fig. 9: It is impossible to read any of the antigens that are 5-fold different even with zooming in and therefore, I cannot appreciate how it would be helpful to the reader. I advise revising the figure so that it is legible.

Response: We appreciate the reviewer's suggestion. We made a new version of the figure and enlarged the font to make it easier to read in updated Fig. 8 (formerly Fig. 9).

Minor concerns:

4. Line 29: specify SARS-CoV-2 "commercial assays"

Response: We detailed the commercial assays in the method section. We added the commercial assays names SARS-CoV-2 chemiluminescent IgG II assay (Beckman), Platelia SARS-CoV-2 total Ab ELISA (Bio-Rad), in line 29-30.

5. Line 84: The authors describe the development of a "clinical testing-compatible" method but do not indicate the turnaround time. I believe readers would be interested in knowing how feasible this method is clinically and therefore describing the time would be beneficial.

Response: We appreciate the reviewer's suggestion. We have added the turnaround time as 24 hours in line 87, which is based on the assay run time. It is important to note that for studies that involve thousands of samples, the time required to manage those samples will depend on the level of automation available in the clinical lab.

6. Describe the sources of the pre-2019 and 2020 samples in the Methods

Response: We appreciate the reviewer's suggestion. We described the detail of the pre-2019 samples and early 2020 samples in the methods section in line 455-459.

“To assess SARS-CoV-2 MISPA assay’s efficacy, there were 64 pre-2019 samples used for evaluation of the efficacy of SARS-CoV-2 analysis that had been collected through a previous unrelated study under the IRB of STUDY00009580. Colleagues at the Mayo Clinic Clinical Testing Laboratory kindly provided 55 qPCR test-confirmed COVID-19 deidentified samples collected between 0-7 days, 8-14 days, and >14 days after symptom onset in early 2020.”

7. *Fig. 7 panels should read "ANOVA"*

Response: We are sorry for the typo, and it was corrected in the updated Fig. 6 (formerly Fig. 7).

8. *Table. 1 should read "Survey"*

Response: We are sorry for the typo, and it was corrected in the updated Table 2 (formerly Table 1).

9. *Lines 570-611: Statistics are presented in the Results and figures that have not been described in the Methods.*

Response: We added the statistics analysis in these method sections, which can be found in the “Marked-Up Manuscript” file in line 632-651.

*“Reproducibility of response against all antigens for 185 samples, the commercial pre-2019 contrived serum sample (Fig. 3D and E) and the response of individual antigens common to PL_147 and PL_184 (Fig. 3F) were assessed through a linear regression model and R-squared value. The seropositivity for each antigen was calculated separately for SurveyFall21 and SurveySpring22 based on the technical cutoffs (5*mean) of the blank control. The Pearson correlation coefficient was used as the distance metric for clustering analysis for SurveyFall21, and the same cluster was used to generate a heatmap of SurveySpring22 (Fig. 4). The correlation clustering patterns across all antigens for two surveys was assessed through Rand Index analysis (figs. S3). To understand the serology responses of coronaviruses resistant subpopulations, the 5th percentile of abundance for each of the seasonal coronaviruses was analyzed by the one-sided Fisher’s exact test (Fig. 5). The antibody differences between self-reported COVID-19 positive and negative (confirmed with Bio-Rad Platelia SARS-CoV-2 total Ab ELISA assay anti-NC negative) was compared by Wilcoxon rank-sum test. Antibody responses across age, gender and race groups using the one-way ANOVA (Fig. 6). To evaluate antibody changes over time, we further analyze the 137 subjects who participated in both surveys, and a 5-fold change in either direction was taken as an arbitrary cutoff for detecting antibody level changes (Fig. 7 and 8). Overall anti-SARS-CoV-2 NC responses between two surveys for 137 common samples were*

compared by a two-tailed paired *t* test (Fig. 7E). The response difference of the SR+ and SR- groups in each survey were compared by Wilcoxon rank-sum test (figs. S4).”

10. Line 581: Address how "equivocal" results were handled for performance calculations

Response: We are sorry for not clear explanation. We rewrite the paragraph for what we did in the method section in line 663-672.

“To evaluate the performance of MISPA compared to qPCR-verified COVID-19 positive and pre-2019 samples, we calculated the Positive Percent Agreement (PPA), Negative Percent Agreement (NPA). The seropositive cutoffs for SARS-CoV-2 wRBD and NC for MISPA were calculated based on the contrived pre-2019 sample (mean+3*SD). The comparison of MISPA and qPCR-verified COVID-19 positivity was calculated based on the following formulas.

- $PPA = (\# \text{ both assays positive}) / (\# \text{ clinical assay positive}) \times 100$
- $NPA = (\# \text{ both assays negative}) / (\# \text{ clinical assay negative}) \times 100$ ”

11. Material and Methods: For proprietary names of assay kits and reagents, include the manufacturer followed by location.

Response: We added the proprietary names of assay kits and reagents, including the manufacturer followed by location.

Reviewer #3 (Comments for the Author):

This manuscript describes a new platform, called Multiplexed In-Solution Protein Array (MISPA), that can analyze blood samples on a large scale. The MISPA assay utilizes folded protein antigens fused with a halo-tag and coupled to DNA barcodes to evaluate antigen-specific antibodies in serum by NGS. The authors claim that they were able to evaluate antibody responses to 39 bacteria and 99 viruses in 2400 individuals using their MISPA assay. While the reported MISPA assay builds upon other successful methods reported previously for detecting immune responses through NGS, this appears to be the first study to demonstrate the assay's potential for evaluating 100+ antigens (147) and 1000+ clinical samples (2400). Overall, the manuscript is well-written and has the potential to advance immune profiling methods for larger sample sizes. However, I have several concerns that must be addressed before publication.

1. The authors mentioned using 147 antigens (39 bacterial and 99 viral proteins) to evaluate the MISPA assay. Could the authors provide the details for designing the antigen panel?

Response: This antigen panel was developed in the summer of 2020 and was designed to understand the serological response for primarily respiratory infections, but to also include other relevant common pathogens. Clearly key areas of focus included SARS-CoV-2, the seasonable coronaviruses, as well common viral and bacterial pathogens of interest. Initial studies in our group that included all coronavirus proteins (whole proteomes from SARS-CoV-2 and all seasonal coronaviruses) showed that RBD and NC were by far the most seroreactive antigens (a result well supported in the literature). Our lab also has considerable experience testing serological responses to many microbial proteins on our protein array platform, from a list of close to 30,000 possible antigens. From these studies, we knew which proteins from various pathogens showed the most prevalent responses in populations. Thus, for the other pathogens of interest, where possible, we selected the most immunodominant antigens based on our previous studies. We added a mark-up explanation in the antigens selection as below in line 164-169.

“The coronavirus antigens (SARS-CoV-2 and seasonal) were selected based on whole proteome studies on our protein microarrays, as well as the literature, that showed RBD and NC were the most seroreactive. We also included many respiratory pathogens, as well as other common pathogens of interest. For these other pathogens, where possible, we included the antigens from each that showed the most prevalent responses in our protein microarray studies.”

2. Additionally, many antigens reported were negative or yielding undetectable signals except certain respiratory viruses and a limited number of pathogenic bacteria. They reported that the samples evaluated were from adults in the ASU community in September 2021 and March 2022, and their seroprevalence estimate matched previous reports. However, it's unclear how the authors confirmed the retained antigenicity of the bacterial and viral antigens upon fusing to the halo tag.

Response: We appreciate the reviewer's insightful comment. As noted, there are some pathogens for which we did not observe any response in this study. This could be because no one in these populations had responses to these organisms or because the antigens in the assay are not good probes for measuring responses to those organisms. Ideally, we would prefer to develop each antigen by evaluating samples from known positive cases and known negative controls to determine the sensitivity and specificity of each antigen. After searching for them, we could not find such sample sets for the overwhelming majority of the organisms listed here. Without clinical positive and negative samples, we cannot directly assess whether the antigen selected is a good proxy for history of infection as well as whether the Halo tag affects its antigenicity in the assay. We are committed to evaluate these antigens more thoroughly by reaching out to the infectious disease community to find better sample sets.

It is important to draw a distinction between antigens that show very low prevalence and those that never show any response. Arguably the latter group raises the concern that those antigens are either inappropriate selections as representative for that infection, or that something in their preparation (e.g., addition of the Halo tag or expression by the IVTT) caused them to lose their antigenicity. We will work further on that relatively small subset to sort that issue out. Some of the antigens here had very low prevalence but still showed positive responses in some individuals. Thus, those proteins retain at least some antigenicity. In some cases, low prevalence makes sense with our “healthy” university population, such as with HIV antigens, p53 (usually a measure of cancer) and the E6 and E7 antigens of oncogenic HPV strains (also found more commonly with cancer). Notably, in other experiments, not included here, we have observed very strong responses to these antigens in relevant populations (AIDS patients and cancer patients). After this analysis, we compared some seroreactive antigens and literature reports and they matched well as stated below (line 222-229).

“There were 37 other microbial antigens (27.0%) that had strong responses in more than 90% of subjects, including SARS-CoV-2 and those from other respiratory viruses (e.g., Human parainfluenza virus 3 (23), Human respiratory syncytial virus (24, 25), seasonal coronavirus (26), Influenza A virus (18) and Influenza B virus, Rhinovirus A (27), and Human mastadenovirus B, C, D (24, 25)), gastrointestinal viruses (e.g., Enterovirus A, B (28)), and pathogenic bacteria (e.g., Staphylococcus aureus (29), Haemophilus influenzae (30, 31), Pseudomonas aeruginosa, and Klebsiella oxytoca), all of which have been reported to have close to 100% seroprevalence in individuals more than 2 years old.”

We also added the following limitation in the discussion section (line 398-407).

“There were 30 proteins for which we did not observe a response ($\leq 5.0\%$ seroprevalence) in this study, which featured a largely healthy population. However, 20 of these proteins showed responses in other studies we have done, particularly studies that included cancer or AIDS patients, suggesting that they are antigenic. There were 10 antigens that have showed low response in all of our studies to date, including antigens from HBV and yellow fever virus. For these 10 antigens, the low response could be because no one in these populations had responses to these organisms or because the antigens in the assay are not good probes for measuring responses to those organisms. We hope to develop each antigen by evaluating samples from known positive cases and known negative controls to determine the sensitivity and specificity of each antigen.”

3. The MISPA assay seems to rely on fusing the antigen with a Halo-tag. Do the authors anticipate that this could pose a problem for the MISPA assay's ability to be more comprehensive, given that many viral glycoproteins may be more difficult to express with Halotag? Additionally, can the authors comment on any concerns about steric hindrance affecting PCR amplification of certain antigens based on their size and shape?

Response: The reviewer raises an important and yet inescapable concern. All serology assays, including ELISA, protein arrays, Luminex beads, MSD, phage display, peptide scans, and lateral flow, rely on attaching the antigen to something in order to read the positive signals. In most cases, the antigens are affixed to a surface, which not only requires an attachment point on the protein, but also limits the degrees of freedom of interaction because the protein is immobilized. We believe that a key advantage of MISPA is that the protein remains in solution allowing it to bind more freely to its interactors. This may be why MISPA was more sensitive in detecting RBD and NC than the commercial assays.

Still, there must always be an attachment point, and this may potentially cause problems, such as with steric hindrance. There are two strategies. One strategy is to chemically, or non-specifically, attach the protein, such as coating a plastic ELISA plate with protein. This has the advantage that attachment points are distributed around the protein, so every face of the protein may get some exposure. But chemical linkage has many disadvantages, including holding the protein very close to the surface (limiting access), often denaturing the protein (losing conformational epitopes) and significantly limiting throughput. The other approach, which we and most others use, is to fuse the gene to an epitope tag. This avoids those limitations, but also brings its own, such as potentially blocking whichever terminus the tag is fused to. A detailed discussion about the benefits of each approach is beyond this response, but we believe the benefits of a fused tag outweigh the limitations. Halo tag has a well-established history and it behaves well as a fusion protein, generally not interfering with most assays. In our side-by-side studies, sensitivity of detection by MISPA outperformed ELISA and our protein microarrays for at least a dozen proteins. This approach is probably sufficient for research purposes where the goal is to measure a positive signal without making statements about the absence of one. However, to bring this to a clinical test, where confidently stating a negative test is often important (e.g., no evidence of infection), we will need to evaluate each antigen in known clinical positive and negative samples.

We agree with the reviewer that additional work may be required for evaluating MISPA for glycoproteins. We will note that we produced the SARS-CoV-2 RBD protein in expi293 cells, which included post translational modification, and MISPA performed comparably to commercial assays for this protein. It is likely that a similar approach would be needed for other proteins that rely on PTM for antigenicity. Each would have to be tested to demonstrate detection of antibodies in serum, as we have done.

Regarding the concern about steric hindrance of PCR amplification by certain antigens based on their size and shape, we note that our design included a polyA linker between the attachment of the linker to the Halo protein and the target DNA barcode to avoid this issue. For the proteins ranging from 50 kDa to 150 kDa in our study, the protein expression level (based on in-gel fluorescence assay) aligned well with the barcoded protein quantification (PCR product and NGS readout). Thus, there do not appear to be significant biases on PCR amplification for the MISPA assay.

4. Can the authors explain why the reported LOD of the MISPA assay (~50 ng/ml) is lower than that of traditional ELISA and orders of magnitude less than multiplex assay, for example, Luminex assay platform? Please discuss this limitation in the manuscript.

Response: We appreciate the reviewer's comments. It is difficult to compare our LOD to other assays without doing a side-by-side comparison. Ultimately each LOD measurement depends on how the experiment was performed, especially the target protein measured, and the antibody used to measure it. Our measured RBD LOD relied on the monoclonal antibody we used to bind RBD, which has its own affinity for the protein and its own behavior in assays. We have noticed in our lab that this antibody performs better in an ELISA format than by MISPA. That said, for other antibodies and antigens that we have tested in the same manner, MISPA outperformed ELISA (and other platforms) in LOD, achieving <1 ng/ml LOD. In this setting, we made no attempts to improve the LOD, either by optimizing this monoclonal antibody or by looking for a better one. Our focus was to develop a test that was comparable to clinical assays and multiplexed. Nonetheless, we agree it worth mentioning this in discussion in line 364-371:

“The LOD for detecting the RBD protein here was less than has been reported by other assays. It is difficult to compare our LOD to other assays without doing a side-by-side comparison. Ultimately each LOD measurement depends on how the experiment was performed, especially the target protein measured. Our measured LOD relied on the monoclonal antibody we used to bind RBD, which has its own affinity for the protein and its own behavior in assays. In these experiments, our focus was to develop an assay that could quantitatively analyze multiple antibodies, and we did not optimize for LOD, which is something that will be needed in the future.”

5. The authors reported that the PPA for NC using MISPA was 90.9%, and the NPA was 98.4% compared to the EUA-approved test. However, a serosurvey in Table 1 showed that the percentage of people who tested positive for COVID-19 and had detectable antibodies increased from 56.9% in September 2021 to 72% in March 2022 using the EUA-approved test. In contrast, the increase in MISPA assay during this period was from 84.3% to 87.5%. Could the authors explain this discrepancy?

Response: We appreciate the reviewer's comments. To assess the positive percent agreement (PPA) and negative percent agreement (NPA), we used samples for which we had categorical assignment evidence, that is, with qPCR-verified COVID-19 positive samples (for positives) and pre-2019 samples (for negatives), respectively. The PPA for NC using MISPA was 90.9% and its NPA was 98.4%. We should note that these samples were categorized as a “clinical validation sample set,” which we received from the Mayo Clinic clinical testing lab. The pre-2019 samples were collected before 2019 and were not expected to be responsive for SARS-CoV-2 NC and the qPCR-verified COVID-19 positive sample were collected during the early 2020 from individuals displaying COVID-19 symptoms. The PPA and NPA we reported here was a proficiency test to verify the SARS-CoV-2 NC's performance in MISPA.

For the serosurvey result in updated Table 2 (formerly Table 1), the SARS-CoV-2 NC in MISPA were tested for SR+/- (self-reported) who reported having had COVID-19, but had not been previously tested for antibodies for SARS-CoV-2 NC. We used the Platelia SARS-CoV-2 total Ab ELISA assay from Bio-Rad to assess their anti-SARS-CoV-2 NC serological reactivity, where the percentage of people who had detectable antibodies was 56.9% in September 2021 and 72% in March 2022. Since **all** of these subjects reported having had a positive test for COVID-19, we would expect nearly all to have anti-NC reactivity. The change from fall to spring is not really an increase in the prevalence (which should be high for both), but rather variation in detection by the assay. For the same set of samples, the MISPA test for anti-SARS-CoV-2 NC, showed a higher prevalence of 84.3% in the first group and 87.5% for the second, which reveals both a better detection of the response and a more consistent one. It is important to note that while both populations self-reported having had COVID-19, we do not know the interval between their case of COVID-19 and when their sample was collected. The timing of serum sample collection, their self-report accuracy, and their vaccination status could affect their anti-SARS-CoV-2 NC response. There are reports that some anti-NC assays wane over time after the infection. What we observed here was that MISPA assay performance was much better than the Platelia SARS-CoV-2 total Ab ELISA for SARS-CoV-2 NC in the SR+ sample group.

6. Several studies have reported Ab waning and affinity maturation following SARS-CoV-2 infection or vaccination. However, the authors of this study conducted two serosurveys involving 137 participants and found that quantitative antibody responses to almost all antigens within each participant remained stable during this period. Notably, the authors tested these samples at a serum dilution of 1:5. Can the authors clarify whether the MISPA assay can accurately measure antibody concentration at this serum dilution? How was the 1:5 serum dilution determined? Since the first serosurvey was conducted using samples collected before the emergence of the Omicron variant (in September 2021), and the second serosurvey was conducted during the Omicron wave (in March 2022), can the authors comment on the difference in antibody reactivity to Wuhan and Omicron antigens, as both these antigens appear to be included in the panel?

Response: The 1:5 serum dilution was based on a Design of Experiment (DOE) optimization that we implemented before the serosurvey study. The serum dilution of 1:5 provided the best serological accuracy for SARS-CoV-2 wRBD and NC based on clinical positive and negative samples (data not shown). These assay conditions allow for enough protein G beads' capacity based on the serum amount to allow all IgG antibodies in the sample to be collected based on the vendor's recommendation. For both serosurveys, our two populations were highly vaccinated (>90% vaccination rates), the majority of the samples produced a signal that appeared to be saturated for anti-SARS-CoV-2 wRBD at the high end of the dynamic range. As their response was out of the linear range of the MISPA assay, the antibody concentration for SARS-CoV-2 wRBD could not be accurately evaluated at that dilution. (We have since determined that further sample dilution of 1:25 gives linear responses for RBD.) However, the anti-NC responses were mainly due to infection and their responses were within the linear

range of the MISPA assay. For 51 of the 137 individuals who participated in both surveys, we observed more than 5-fold increase of anti-SARS-CoV-2 NC (Fig. 7E), the majority of them (41; 80.4%) reported having COVID-19 between the two surveys. We also did a serial dilution of a limited set of samples and 1:5 serum dilution had shown that the majority of the antigens were within the dynamic response range (data not shown).

We included a supplementary Fig. S7 that shows signal for SARS-CoV-2 antigens across all 137 participants. As noted above and evidenced by the dramatic increase in anti-NC activity, more than a third of these participants had infections between the fall (2021) and spring (2022) serosurveys – presumably most of these were Omicron. As we noted, signal for the Wuhan protein was so strong that the signal was nearly saturated, whereas the anti-Omicron signal was much weaker. We and others find it much harder to produce and purify the Omicron RBD protein. Nevertheless, we did observe a strong increase in reactivity to the Omicron RBD from fall to spring that correlates well with the many new cases during that window. We added this result in the result as well (line 328-334).

“We specifically looked at differences in response to the SARS-CoV-2 antigens for all the 137 participants. As expected, all three, the SARS-CoV-2 wRBD, oRBD, and NC showed significant increases (figs. S7) using the RankSum test, consistent with known new cases during that time window. The SARS-CoV-2 oRBD showed the highest significance ($p < 0.0001$), which agreed with the Omicron wave in early 2022, though notably the signal strength for that protein is weaker than for the wRBD. Around half of the population had anti- SARS-CoV-2 NC increased as observed in Fig. 7E.”

7. Numerous studies have reported the correlation between measured SARS-CoV-2 RBD binding Abs and SARS-CoV-2 neutralization titer. Have the authors tried correlating the RBD binding signal from the MISPA assay to SARS-CoV-2 neutralization titers?

Response: We are interested to understand the correlation between the SARS-CoV-2 RBD binding Abs and the SARS-CoV-2 neutralization titer. However, we haven't finished the development of an assay for SARS-CoV-2 neutralization yet. For the serosurvey samples, we did not have neutralization titer information for us to compare to. We are planning to continue the development of a neutralization assay through MISPA. We added this comment in our discussion (line 423-426).

“It would be interesting to also understand the correlation between the SARS-CoV-2 RBD binding Abs and the SARS-CoV-2 neutralization titer. The implementation of a neutralization antibody assay based on competition with their binding with specific antigen would be another possible use of MISPA.”

8. Have the authors tried to identify IgM or IgA antibodies? It is described that the protein G beads were used to pull the immunocomplex, which will be selective for IgG Abs. Please discuss the potential utility of MISPA assay to detect other Ab isotypes.

Response: We appreciated the reviewer's suggestions. In these studies, for speed of development, we used protein G to capture IgG pair after a careful test of the Protein G magnetic beads through various vendors. That said, there is nothing precluding the use of anti-IgM and -IgA antibodies for antibody capture and we are trying those strategies. We added this comment in our discussion (420-423)

"The current MISPA assay was based on the profiling of IgG antibodies through Protein G-IgG capture. It would be straightforward to also evaluate IgA and IgM antibodies, which can be performed with the appropriate capture system to profile their response along with IgG, to empower the understanding of the immune response in a further depth."

9. In lines 164-165, it was stated that the MISPA assay dynamic range for wRBD was from 38.9-100,000 ng/mL of anti-RBD (Fig. 1B). However, data presented in Fig 1B appear to saturate around 10,000. Should the dynamic range be between 38.9 and 10,000?

Response: We do notice that the response of 100,000 ng/ml was close to that of 10,000 ng/ml and was not in the linear range for the concentration-response curve. However, as the Y-axis is in log range where a change could still be reproducibly detected. Hence, we assessed the signal dynamic range (as opposed to the linear response range) was from 38.9-100,000 ng/ml.

10. In line 464, provide the catalog number if Azido chloroalkane is commercially available. Otherwise, please provide details of its synthesis for reproducibility.

Response: No. we performed a customized synthesis to produce the material at high yield. However, the material itself is the same as the catalog, RL-3710 from Iris biotech. The name is Halo-PEG(4)-Azide, chemical name is 1-Azido-18-chloro-3,6,9,12-tetraoxaocadecane (C₁₄H₂₈ClN₃O₄) and the mass size is 337,85 g/mol. We included this in the methods section (line 511-514).

11. In line 487, it was mentioned that 20 µl of the expressed protein was used for halo-tagged protein barcoding. However, considering that expression levels may vary across constructs, giving an estimated range of the protein quantity used in this reaction would be best. How the conjugation efficiency of chloroalkane with the halo-tagged protein was evaluated?

Response: Yes, the expression level will vary across different proteins. The protein expression level from the IVTT we used expresses up to 100 µg/ml based on the

vendor, which was up to 1-2 μM for an average size of 50-100 kDa protein. The concentration of the Halo ligand we used for protein barcoding was 10 μM with the same volume amount. Hence, we allow the excess of Halo ligand to allow all proteins to be fully barcoded before anti-flag beads purification. We added this information to the method section.

12. In lines 490-491, provide the source and catalog number for anti-flag-coated magnetic beads.r #1:

Response: We added the information for the anti-flag-coated magnetic beads.

Re: Spectrum02399-23R1 (Quantitative assessment of multiple pathogen exposure and immune dynamics at scale)

Dear Prof. Joshua LaBaer:

Thank you for the privilege of reviewing your work. Below you will find my comments, instructions from the Spectrum editorial office, and the reviewer comments.

I strongly recommend moving Figure 8 into Supplemental Figures. There are already a lot of figures in the manuscript, Figure 8 is quite large with 4 full-page panels, and it's not described in great detail in the text.

Revision Guidelines

Sincerely,
Kileen Shier
Editor
Microbiology Spectrum

Response to the Reviewers' comments:

Reviewer #1 (Comments for the Author):

Thank you for the opportunity to review the manuscript "Quantitative Assessment. Of multiple pathogen exposure and immune dynamics at scale". The manuscript describes the use of a multiplexed serology method, MISPA, to evaluate the serological response using a large-scale, high-throughput method with the ability to assess for antigens to 39 bacteria and 99 viruses from a single sample. The method was evaluated in 2400 individuals for two surveillance time periods during the COVID-19 pandemic. The method was found to have high sensitivity, a wide dynamic range and performs favorably compared to commercially available SARS-COV-2 serology assays. In addition, this method captured quantitative longitudinal stability of the serological responses over the time period between the two surveys that could be correlated with new infections or vaccination events highlighting the potential use for the method to be applied as an epidemiology tool. The manuscript presents a comprehensive overview of the results and methods, making it an intriguing read that would capture the interest of its readers.

Major Concerns:

1. Results section: This section could be streamlined to just results. There are some instances of information that might be better suited for the Methods and other instances of discussion of the results (eg. Lines 322-330) that make more sense to be in the discussion section.

Response: We appreciate the reviewer's kind suggestion. We removed some of the information to the methods section or the discussion section to make the results section more streamlined. The changes were marked in blue in the "Marked-Up Manuscript" file.

We removed the "A modified oligonucleotide, 5' 5-Octadiynyl dU modified DNA, was conjugated to the chloroalkane through Cu(I)-catalyzed azide-alkyne cycloaddition" in result section (line 109-110).

We moved the "which has been employed extensively in the production of tens of thousands of proteins for use in serological studies (13-19). This HeLa cell lysate-based IVTT, which includes both human ribosomes and chaperone proteins, can produce folded and enzymatically active proteins (20). As extracellular domains and proteins reside in a different biochemical environment, the IVTT does not always yield well-folded extracellular proteins, perhaps due to the lack of disulfide bonds and other post translational modifications (PTM). Thus, we produced some protein targets using" in the result section to the discussion section (line 411-419).

We moved the "The constant level of antibody responses over a 6-month time window is much longer than the half-life of antibodies (< 4 weeks (38)). This quantitative stability against so many antigens agrees with the known durability of vaccine responses (39) and implies some form of feedback regulation that maintains specific antibodies at a fixed level in the blood over time. This regulation may occur at the level of plasma cells, which are known to survive for longer than a year or may include a more complex mechanism including memory B cells (39-43). Understanding the factors

responsible for setting antibody levels, as well as the mechanisms for maintaining them, will contribute to improved disease tracking and offer the potential for immune adjustment.” to the discussion section (line 377-385).

2. Consider moving Fig. 3 to supplemental and moving the individual demographics Table S2 to the main text. Since there are subanalyses performed (vaccination seropositivity, age, gender etc.) that are addressed in the main text and figures it would be easier for readers to refer to the table in the main article.

Response: We appreciate the reviewer's suggestion. We moved Fig. 3 to figs. S2 and the individual demographics Table S2 to the main text as Table 1.

3. Fig. 9: It is impossible to read any of the antigens that are 5-fold different even with zooming in and therefore, I cannot appreciate how it would be helpful to the reader. I advise revising the figure so that it is legible.

Response: We appreciate the reviewer's suggestion. We made a new version of the figure and enlarged the font to make it easier to read in updated figs. S7 (formerly Fig. 9).

Minor concerns:

4. Line 29: specify SARS-CoV-2 "commercial assays"

Response: We detailed the commercial assays in the method section. We added the commercial assays names SARS-CoV-2 chemiluminescent IgG II assay (Beckman), Platelia SARS-CoV-2 total Ab ELISA (Bio-Rad), in line 29-30.

5. Line 84: The authors describe the development of a "clinical testing-compatible" method but do not indicate the turnaround time. I believe readers would be interested in knowing how feasible this method is clinically and therefore describing the time would be beneficial.

Response: We appreciate the reviewer's suggestion. We have added the turnaround time as 24 hours in line 87, which is based on the assay run time. It is important to note that for studies that involve thousands of samples, the time required to manage those samples will depend on the level of automation available in the clinical lab.

6. Describe the sources of the pre-2019 and 202 samples in the Methods

Response: We appreciate the reviewer's suggestion. We described the detail of the pre-2019 samples and early 2020 samples in the methods section in line 455-459.

“To assess SARS-CoV-2 MISPA assay’s efficacy, there were 64 pre-2019 samples used for evaluation of the efficacy of SARS-CoV-2 analysis that had been collected through a previous unrelated study under the IRB of STUDY00009580. Colleagues at the Mayo Clinic Clinical Testing Laboratory kindly provided 55 qPCR test-confirmed COVID-19 deidentified samples collected between 0-7 days, 8-14 days, and >14 days after symptom onset in early 2020.”

7. *Fig. 7 panels should read "ANOVA"*

Response: We are sorry for the typo, and it was corrected in the updated Fig. 6 (formerly Fig. 7).

8. *Table. 1 should read "Survey"*

Response: We are sorry for the typo, and it was corrected in the updated Table 2 (formerly Table 1).

9. *Lines 570-611: Statistics are presented in the Results and figures that have not been described in the Methods.*

Response: We added the statistics analysis in these method sections, which can be found in the “Marked-Up Manuscript” file in line 632-651.

*“Reproducibility of response against all antigens for 185 samples, the commercial pre-2019 contrived serum sample (Fig. 3D and E) and the response of individual antigens common to PL_147 and PL_184 (Fig. 3F) were assessed through a linear regression model and R-squared value. The seropositivity for each antigen was calculated separately for SurveyFall21 and SurveySpring22 based on the technical cutoffs (5*mean) of the blank control. The Pearson correlation coefficient was used as the distance metric for clustering analysis for SurveyFall21, and the same cluster was used to generate a heatmap of SurveySpring22 (Fig. 4). The correlation clustering patterns across all antigens for two surveys was assessed through Rand Index analysis (figs. S3). To understand the serology responses of coronaviruses resistant subpopulations, the 5th percentile of abundance for each of the seasonal coronaviruses was analyzed by the one-sided Fisher’s exact test (Fig. 5). The antibody differences between self-reported COVID-19 positive and negative (confirmed with Bio-Rad Platelia SARS-CoV-2 total Ab ELISA assay anti-NC negative) was compared by Wilcoxon rank-sum test. Antibody responses across age, gender and race groups using the one-way ANOVA (Fig. 6). To evaluate antibody changes over time, we further analyze the 137 subjects who participated in both surveys, and a 5-fold change in either direction was taken as an arbitrary cutoff for detecting antibody level changes (Fig. 7 and figs. S7). Overall anti-SARS-CoV-2 NC responses between two surveys for 137 common samples were*

compared by a two-tailed paired *t* test (Fig. 7E). The response difference of the SR+ and SR- groups in each survey were compared by Wilcoxon rank-sum test (figs. S4).”

10. Line 581: Address how "equivocal" results were handled for performance calculations

Response: We are sorry for not clear explanation. We rewrite the paragraph for what we did in the method section in line 663-672.

“To evaluate the performance of MISPA compared to qPCR-verified COVID-19 positive and pre-2019 samples, we calculated the Positive Percent Agreement (PPA), Negative Percent Agreement (NPA). The seropositive cutoffs for SARS-CoV-2 wRBD and NC for MISPA were calculated based on the contrived pre-2019 sample (mean+3*SD). The comparison of MISPA and qPCR-verified COVID-19 positivity was calculated based on the following formulas.

- $PPA = (\# \text{ both assays positive}) / (\# \text{ clinical assay positive}) \times 100$
- $NPA = (\# \text{ both assays negative}) / (\# \text{ clinical assay negative}) \times 100$ ”

11. Material and Methods: For proprietary names of assay kits and reagents, include the manufacturer followed by location.

Response: We added the proprietary names of assay kits and reagents, including the manufacturer followed by location.

Reviewer #3 (Comments for the Author):

This manuscript describes a new platform, called Multiplexed In-Solution Protein Array (MISPA), that can analyze blood samples on a large scale. The MISPA assay utilizes folded protein antigens fused with a halo-tag and coupled to DNA barcodes to evaluate antigen-specific antibodies in serum by NGS. The authors claim that they were able to evaluate antibody responses to 39 bacteria and 99 viruses in 2400 individuals using their MISPA assay. While the reported MISPA assay builds upon other successful methods reported previously for detecting immune responses through NGS, this appears to be the first study to demonstrate the assay's potential for evaluating 100+ antigens (147) and 1000+ clinical samples (2400). Overall, the manuscript is well-written and has the potential to advance immune profiling methods for larger sample sizes. However, I have several concerns that must be addressed before publication.

1. The authors mentioned using 147 antigens (39 bacterial and 99 viral proteins) to evaluate the MISPA assay. Could the authors provide the details for designing the antigen panel?

Response: This antigen panel was developed in the summer of 2020 and was designed to understand the serological response for primarily respiratory infections, but to also include other relevant common pathogens. Clearly key areas of focus included SARS-CoV-2, the seasonable coronaviruses, as well common viral and bacterial pathogens of interest. Initial studies in our group that included all coronavirus proteins (whole proteomes from SARS-CoV-2 and all seasonal coronaviruses) showed that RBD and NC were by far the most seroreactive antigens (a result well supported in the literature). Our lab also has considerable experience testing serological responses to many microbial proteins on our protein array platform, from a list of close to 30,000 possible antigens. From these studies, we knew which proteins from various pathogens showed the most prevalent responses in populations. Thus, for the other pathogens of interest, where possible, we selected the most immunodominant antigens based on our previous studies. We added a mark-up explanation in the antigens selection as below in line 164-169.

“The coronavirus antigens (SARS-CoV-2 and seasonal) were selected based on whole proteome studies on our protein microarrays, as well as the literature, that showed RBD and NC were the most seroreactive. We also included many respiratory pathogens, as well as other common pathogens of interest. For these other pathogens, where possible, we included the antigens from each that showed the most prevalent responses in our protein microarray studies.”

2. Additionally, many antigens reported were negative or yielding undetectable signals except certain respiratory viruses and a limited number of pathogenic bacteria. They reported that the samples evaluated were from adults in the ASU community in September 2021 and March 2022, and their seroprevalence estimate matched previous reports. However, it's unclear how the authors confirmed the retained antigenicity of the bacterial and viral antigens upon fusing to the halo tag.

Response: We appreciate the reviewer's insightful comment. As noted, there are some pathogens for which we did not observe any response in this study. This could be because no one in these populations had responses to these organisms or because the antigens in the assay are not good probes for measuring responses to those organisms. Ideally, we would prefer to develop each antigen by evaluating samples from known positive cases and known negative controls to determine the sensitivity and specificity of each antigen. After searching for them, we could not find such sample sets for the overwhelming majority of the organisms listed here. Without clinical positive and negative samples, we cannot directly assess whether the antigen selected is a good proxy for history of infection as well as whether the Halo tag affects its antigenicity in the assay. We are committed to evaluate these antigens more thoroughly by reaching out to the infectious disease community to find better sample sets.

It is important to draw a distinction between antigens that show very low prevalence and those that never show any response. Arguably the latter group raises the concern that those antigens are either inappropriate selections as representative for that infection, or that something in their preparation (e.g., addition of the Halo tag or expression by the IVTT) caused them to lose their antigenicity. We will work further on that relatively small subset to sort that issue out. Some of the antigens here had very low prevalence but still showed positive responses in some individuals. Thus, those proteins retain at least some antigenicity. In some cases, low prevalence makes sense with our “healthy” university population, such as with HIV antigens, p53 (usually a measure of cancer) and the E6 and E7 antigens of oncogenic HPV strains (also found more commonly with cancer). Notably, in other experiments, not included here, we have observed very strong responses to these antigens in relevant populations (AIDS patients and cancer patients). After this analysis, we compared some seroreactive antigens and literature reports and they matched well as stated below (line 222-229).

“There were 37 other microbial antigens (27.0%) that had strong responses in more than 90% of subjects, including SARS-CoV-2 and those from other respiratory viruses (e.g., Human parainfluenza virus 3 (23), Human respiratory syncytial virus (24, 25), seasonal coronavirus (26), Influenza A virus (18) and Influenza B virus, Rhinovirus A (27), and Human mastadenovirus B, C, D (24, 25)), gastrointestinal viruses (e.g., Enterovirus A, B (28)), and pathogenic bacteria (e.g., Staphylococcus aureus (29), Haemophilus influenzae (30, 31), Pseudomonas aeruginosa, and Klebsiella oxytoca), all of which have been reported to have close to 100% seroprevalence in individuals more than 2 years old.”

We also added the following limitation in the discussion section (line 398-407).

“There were 30 proteins for which we did not observe a response ($\leq 5.0\%$ seroprevalence) in this study, which featured a largely healthy population. However, 20 of these proteins showed responses in other studies we have done, particularly studies that included cancer or AIDS patients, suggesting that they are antigenic. There were 10 antigens that have showed low response in all of our studies to date, including antigens from HBV and yellow fever virus. For these 10 antigens, the low response could be because no one in these populations had responses to these organisms or because the antigens in the assay are not good probes for measuring responses to those organisms. We hope to develop each antigen by evaluating samples from known positive cases and known negative controls to determine the sensitivity and specificity of each antigen.”

3. The MISPA assay seems to rely on fusing the antigen with a Halo-tag. Do the authors anticipate that this could pose a problem for the MISPA assay's ability to be more comprehensive, given that many viral glycoproteins may be more difficult to express with Halotag? Additionally, can the authors comment on any concerns about steric hindrance affecting PCR amplification of certain antigens based on their size and shape?

Response: The reviewer raises an important and yet inescapable concern. All serology assays, including ELISA, protein arrays, Luminex beads, MSD, phage display, peptide scans, and lateral flow, rely on attaching the antigen to something in order to read the positive signals. In most cases, the antigens are affixed to a surface, which not only requires an attachment point on the protein, but also limits the degrees of freedom of interaction because the protein is immobilized. We believe that a key advantage of MISPA is that the protein remains in solution allowing it to bind more freely to its interactors. This may be why MISPA was more sensitive in detecting RBD and NC than the commercial assays.

Still, there must always be an attachment point, and this may potentially cause problems, such as with steric hindrance. There are two strategies. One strategy is to chemically, or non-specifically, attach the protein, such as coating a plastic ELISA plate with protein. This has the advantage that attachment points are distributed around the protein, so every face of the protein may get some exposure. But chemical linkage has many disadvantages, including holding the protein very close to the surface (limiting access), often denaturing the protein (losing conformational epitopes) and significantly limiting throughput. The other approach, which we and most others use, is to fuse the gene to an epitope tag. This avoids those limitations, but also brings its own, such as potentially blocking whichever terminus the tag is fused to. A detailed discussion about the benefits of each approach is beyond this response, but we believe the benefits of a fused tag outweigh the limitations. Halo tag has a well-established history and it behaves well as a fusion protein, generally not interfering with most assays. In our side-by-side studies, sensitivity of detection by MISPA outperformed ELISA and our protein microarrays for at least a dozen proteins. This approach is probably sufficient for research purposes where the goal is to measure a positive signal without making statements about the absence of one. However, to bring this to a clinical test, where confidently stating a negative test is often important (e.g., no evidence of infection), we will need to evaluate each antigen in known clinical positive and negative samples.

We agree with the reviewer that additional work may be required for evaluating MISPA for glycoproteins. We will note that we produced the SARS-CoV-2 RBD protein in expi293 cells, which included post translational modification, and MISPA performed comparably to commercial assays for this protein. It is likely that a similar approach would be needed for other proteins that rely on PTM for antigenicity. Each would have to be tested to demonstrate detection of antibodies in serum, as we have done.

Regarding the concern about steric hindrance of PCR amplification by certain antigens based on their size and shape, we note that our design included a polyA linker between the attachment of the linker to the Halo protein and the target DNA barcode to avoid this issue. For the proteins ranging from 50 kDa to 150 kDa in our study, the protein expression level (based on in-gel fluorescence assay) aligned well with the barcoded protein quantification (PCR product and NGS readout). Thus, there do not appear to be significant biases on PCR amplification for the MISPA assay.

4. Can the authors explain why the reported LOD of the MISPA assay (~50 ng/ml) is lower than that of traditional ELISA and orders of magnitude less than multiplex assay, for example, Luminex assay platform? Please discuss this limitation in the manuscript.

Response: We appreciate the reviewer's comments. It is difficult to compare our LOD to other assays without doing a side-by-side comparison. Ultimately each LOD measurement depends on how the experiment was performed, especially the target protein measured, and the antibody used to measure it. Our measured RBD LOD relied on the monoclonal antibody we used to bind RBD, which has its own affinity for the protein and its own behavior in assays. We have noticed in our lab that this antibody performs better in an ELISA format than by MISPA. That said, for other antibodies and antigens that we have tested in the same manner, MISPA outperformed ELISA (and other platforms) in LOD, achieving <1 ng/ml LOD. In this setting, we made no attempts to improve the LOD, either by optimizing this monoclonal antibody or by looking for a better one. Our focus was to develop a test that was comparable to clinical assays and multiplexed. Nonetheless, we agree it worth mentioning this in discussion in line 364-371:

“The LOD for detecting the RBD protein here was less than has been reported by other assays. It is difficult to compare our LOD to other assays without doing a side-by-side comparison. Ultimately each LOD measurement depends on how the experiment was performed, especially the target protein measured. Our measured LOD relied on the monoclonal antibody we used to bind RBD, which has its own affinity for the protein and its own behavior in assays. In these experiments, our focus was to develop an assay that could quantitatively analyze multiple antibodies, and we did not optimize for LOD, which is something that will be needed in the future.”

5. The authors reported that the PPA for NC using MISPA was 90.9%, and the NPA was 98.4% compared to the EUA-approved test. However, a serosurvey in Table 1 showed that the percentage of people who tested positive for COVID-19 and had detectable antibodies increased from 56.9% in September 2021 to 72% in March 2022 using the EUA-approved test. In contrast, the increase in MISPA assay during this period was from 84.3% to 87.5%. Could the authors explain this discrepancy?

Response: We appreciate the reviewer's comments. To assess the positive percent agreement (PPA) and negative percent agreement (NPA), we used samples for which we had categorical assignment evidence, that is, with qPCR-verified COVID-19 positive samples (for positives) and pre-2019 samples (for negatives), respectively. The PPA for NC using MISPA was 90.9% and its NPA was 98.4%. We should note that these samples were categorized as a “clinical validation sample set,” which we received from the Mayo Clinic clinical testing lab. The pre-2019 samples were collected before 2019 and were not expected to be responsive for SARS-CoV-2 NC and the qPCR-verified COVID-19 positive sample were collected during the early 2020 from individuals displaying COVID-19 symptoms. The PPA and NPA we reported here was a proficiency test to verify the SARS-CoV-2 NC's performance in MISPA.

For the serosurvey result in updated Table 2 (formerly Table 1), the SARS-CoV-2 NC in MISPA were tested for SR+/- (self-reported) who reported having had COVID-19, but had not been previously tested for antibodies for SARS-CoV-2 NC. We used the Platelia SARS-CoV-2 total Ab ELISA assay from Bio-Rad to assess their anti-SARS-CoV-2 NC serological reactivity, where the percentage of people who had detectable antibodies was 56.9% in September 2021 and 72% in March 2022. Since **all** of these subjects reported having had a positive test for COVID-19, we would expect nearly all to have anti-NC reactivity. The change from fall to spring is not really an increase in the prevalence (which should be high for both), but rather variation in detection by the assay. For the same set of samples, the MISPA test for anti-SARS-CoV-2 NC, showed a higher prevalence of 84.3% in the first group and 87.5% for the second, which reveals both a better detection of the response and a more consistent one. It is important to note that while both populations self-reported having had COVID-19, we do not know the interval between their case of COVID-19 and when their sample was collected. The timing of serum sample collection, their self-report accuracy, and their vaccination status could affect their anti-SARS-CoV-2 NC response. There are reports that some anti-NC assays wane over time after the infection. What we observed here was that MISPA assay performance was much better than the Platelia SARS-CoV-2 total Ab ELISA for SARS-CoV-2 NC in the SR+ sample group.

6. Several studies have reported Ab waning and affinity maturation following SARS-CoV-2 infection or vaccination. However, the authors of this study conducted two serosurveys involving 137 participants and found that quantitative antibody responses to almost all antigens within each participant remained stable during this period. Notably, the authors tested these samples at a serum dilution of 1:5. Can the authors clarify whether the MISPA assay can accurately measure antibody concentration at this serum dilution? How was the 1:5 serum dilution determined? Since the first serosurvey was conducted using samples collected before the emergence of the Omicron variant (in September 2021), and the second serosurvey was conducted during the Omicron wave (in March 2022), can the authors comment on the difference in antibody reactivity to Wuhan and Omicron antigens, as both these antigens appear to be included in the panel?

Response: The 1:5 serum dilution was based on a Design of Experiment (DOE) optimization that we implemented before the serosurvey study. The serum dilution of 1:5 provided the best serological accuracy for SARS-CoV-2 wRBD and NC based on clinical positive and negative samples (data not shown). These assay conditions allow for enough protein G beads' capacity based on the serum amount to allow all IgG antibodies in the sample to be collected based on the vendor's recommendation. For both serosurveys, our two populations were highly vaccinated (>90% vaccination rates), the majority of the samples produced a signal that appeared to be saturated for anti-SARS-CoV-2 wRBD at the high end of the dynamic range. As their response was out of the linear range of the MISPA assay, the antibody concentration for SARS-CoV-2 wRBD could not be accurately evaluated at that dilution. (We have since determined that further sample dilution of 1:25 gives linear responses for RBD.) However, the anti-NC responses were mainly due to infection and their responses were within the linear

range of the MISPA assay. For 51 of the 137 individuals who participated in both surveys, we observed more than 5-fold increase of anti-SARS-CoV-2 NC (Fig. 7E), the majority of them (41; 80.4%) reported having COVID-19 between the two surveys. We also did a serial dilution of a limited set of samples and 1:5 serum dilution had shown that the majority of the antigens were within the dynamic response range (data not shown).

We included a supplementary figs. S8 that shows signal for SARS-CoV-2 antigens across all 137 participants. As noted above and evidenced by the dramatic increase in anti-NC activity, more than a third of these participants had infections between the fall (2021) and spring (2022) serosurveys – presumably most of these were Omicron. As we noted, signal for the Wuhan protein was so strong that the signal was nearly saturated, whereas the anti-Omicron signal was much weaker. We and others find it much harder to produce and purify the Omicron RBD protein. Nevertheless, we did observe a strong increase in reactivity to the Omicron RBD from fall to spring that correlates well with the many new cases during that window. We added this result in the result as well (line 328-334).

“We specifically looked at differences in response to the SARS-CoV-2 antigens for all the 137 participants. As expected, all three, the SARS-CoV-2 wRBD, oRBD, and NC showed significant increases (figs. S8) using the RankSum test, consistent with known new cases during that time window. The SARS-CoV-2 oRBD showed the highest significance ($p < 0.0001$), which agreed with the Omicron wave in early 2022, though notably the signal strength for that protein is weaker than for the wRBD. Around half of the population had anti- SARS-CoV-2 NC increased as observed in Fig. 7E.”

7. Numerous studies have reported the correlation between measured SARS-CoV-2 RBD binding Abs and SARS-CoV-2 neutralization titer. Have the authors tried correlating the RBD binding signal from the MISPA assay to SARS-CoV-2 neutralization titers?

Response: We are interested to understand the correlation between the SARS-CoV-2 RBD binding Abs and the SARS-CoV-2 neutralization titer. However, we haven't finished the development of an assay for SARS-CoV-2 neutralization yet. For the serosurvey samples, we did not have neutralization titer information for us to compare to. We are planning to continue the development of a neutralization assay through MISPA. We added this comment in our discussion (line 423-426).

“It would be interesting to also understand the correlation between the SARS-CoV-2 RBD binding Abs and the SARS-CoV-2 neutralization titer. The implementation of a neutralization antibody assay based on competition with their binding with specific antigen would be another possible use of MISPA.”

8. Have the authors tried to identify IgM or IgA antibodies? It is described that the protein G beads were used to pull the immunocomplex, which will be selective for IgG Abs. Please discuss the potential utility of MISPA assay to detect other Ab isotypes.

Response: We appreciated the reviewer's suggestions. In these studies, for speed of development, we used protein G to capture IgG pair after a careful test of the Protein G magnetic beads through various vendors. That said, there is nothing precluding the use of anti-IgM and -IgA antibodies for antibody capture and we are trying those strategies. We added this comment in our discussion (420-423)

"The current MISPA assay was based on the profiling of IgG antibodies through Protein G-IgG capture. It would be straightforward to also evaluate IgA and IgM antibodies, which can be performed with the appropriate capture system to profile their response along with IgG, to empower the understanding of the immune response in a further depth."

9. In lines 164-165, it was stated that the MISPA assay dynamic range for wRBD was from 38.9-100,000 ng/mL of anti-RBD (Fig. 1B). However, data presented in Fig 1B appear to saturate around 10,000. Should the dynamic range be between 38.9 and 10,000?

Response: We do notice that the response of 100,000 ng/ml was close to that of 10,000 ng/ml and was not in the linear range for the concentration-response curve. However, as the Y-axis is in log range where a change could still be reproducibly detected. Hence, we assessed the signal dynamic range (as opposed to the linear response range) was from 38.9-100,000 ng/ml.

10. In line 464, provide the catalog number if Azido chloroalkane is commercially available. Otherwise, please provide details of its synthesis for reproducibility.

Response: No. we performed a customized synthesis to produce the material at high yield. However, the material itself is the same as the catalog, RL-3710 from Iris biotech. The name is Halo-PEG(4)-Azide, chemical name is 1-Azido-18-chloro-3,6,9,12-tetraoxaocadecane (C₁₄H₂₈ClN₃O₄) and the mass size is 337,85 g/mol. We included this in the methods section (line 511-514).

11. In line 487, it was mentioned that 20 µl of the expressed protein was used for halo-tagged protein barcoding. However, considering that expression levels may vary across constructs, giving an estimated range of the protein quantity used in this reaction would be best. How the conjugation efficiency of chloroalkane with the halo-tagged protein was evaluated?

Response: Yes, the expression level will vary across different proteins. The protein expression level from the IVTT we used expresses up to 100 µg/ml based on the

vendor, which was up to 1-2 μM for an average size of 50-100 kDa protein. The concentration of the Halo ligand we used for protein barcoding was 10 μM with the same volume amount. Hence, we allow the excess of Halo ligand to allow all proteins to be fully barcoded before anti-flag beads purification. We added this information to the method section.

12. In lines 490-491, provide the source and catalog number for anti-flag-coated magnetic beads.r #1:

Response: We added the information for the anti-flag-coated magnetic beads.

Re: Spectrum02399-23R2 (Quantitative assessment of multiple pathogen exposure and immune dynamics at scale)

Dear Prof. Joshua LaBaer:

Your manuscript has been accepted, and I am forwarding it to the ASM production staff for publication. Your paper will first be checked to make sure all elements meet the technical requirements. ASM staff will contact you if anything needs to be revised before copyediting and production can begin. Otherwise, you will be notified when your proofs are ready to be viewed.

Sincerely,
Kileen Shier
Editor
Microbiology Spectrum